# Structural heterogeneity of α-synuclein fibrils amplified from patient brain extracts

Timo Strohäker[1,2], Byung Chul Jung[3], Shu-Hao Liou [4], Claudio O. Fernandez[5,6], Dietmar Riedel[7], Stefan Becker [2], Glenda M. Halliday [8,9], Marina Bennati[4], Woojin S. Kim [8,9]*, Seung-Jae Lee[3]* & Markus Zweckstetter [1,2,10]*

Parkinson's disease (PD) and Multiple System Atrophy (MSA) are clinically distinctive diseases that feature a common neuropathological hallmark of aggregated α-synuclein. Little is known about how differences in α-synuclein aggregate structure affect disease phenotype. Here, we amplified α-synuclein aggregates from PD and MSA brain extracts and analyzed the conformational properties using fluorescent probes, NMR spectroscopy and electron paramagnetic resonance. We also generated and analyzed several in vitro α-synuclein polymorphs. We found that brain-derived α-synuclein fibrils were structurally different to all of the in vitro polymorphs analyzed. Importantly, there was a greater structural heterogeneity among α-synuclein fibrils from the PD brain compared to those from the MSA brain, possibly reflecting on the greater variability of disease phenotypes evident in PD. Our findings have significant ramifications for the use of non-brain-derived α-synuclein fibrils in PD and MSA studies, and raise important questions regarding the *one disease-one strain* hypothesis in the study of α-synucleinopathies.

[1] German Center for Neurodegenerative Diseases (DZNE), Von-Siebold-Str. 3a, 37075 Göttingen, Germany. [2] Department for NMR-based Structural Biology, Max Planck Institute for Biophysical Chemistry, Am Faßberg 11, 37077 Göttingen, Germany. [3] Department of Biomedical Sciences, Neuroscience Research Institute, College of Medicine, Seoul National University, Seoul 03080, Korea. [4] RG EPR Spectroscopy, Max Planck Institute for Biophysical Chemistry, Am Faßberg 11, 37077 Göttingen, Germany. [5] Max Planck Laboratory for Structural Biology, Chemistry and Molecular Biophysics of Rosario (MPLbioR, UNR-MPIbpC), S2002LRK Rosario, Argentina. [6] Instituto de Investigaciones para el Descubrimiento de Fármacos de Rosario (IIDEFAR, UNR-CONICET), Universidad Nacional de Rosario, S2002LRK Rosario, Argentina. [7] Department for Electron Microscopy, Max Planck Institute for Biophysical Chemistry, Am Faßberg 11, 37077 Göttingen, Germany. [8] Brain and Mind Centre and Central Clinical School, Faculty of Medicine and Health, The University of Sydney, Camperdown, NSW 2050, Australia. [9] School of Medical Sciences, University of New South Wales & Neuroscience Research Australia, Randwick, NSW 2031, Australia. [10] Department of Neurology, University Medical Center Göttingen, University of Göttingen, Waldweg 33, 37073 Göttingen, Germany. *email: woojin.kim@sydney.edu.au; sjlee66@snu.ac.kr; Markus.Zweckstetter@dzne.de

The pathological hallmark of neurodegenerative disorders is brain inclusions that contain insoluble proteins[1]. Aggregates of these proteins spread through the brain and seed misfolding and aggregation of protein in host brain cells[1]. Parkinson's disease (PD) and multiple system atrophy (MSA) are characterized by the pathological accumulation of the misfolded protein, α-synuclein, but have different clinical presentations[2]. The clinical differences are being explained in analogy to prion diseases by different α-synuclein aggregate structures[1,3,4]. Consistent with this hypothesis, α-synuclein fibrils with distinct conformations produced in the test tube differ in their cellular seeding properties and toxicity[5,6].

The morphology of insoluble α-synuclein (aSyn) pathologies varies in PD and MSA[2]. In PD, round perikaryal aSyn inclusions termed classical Lewy bodies are present in neurons of the substantia nigra as well as other brain regions[2]. In addition, less structured fibrillar inclusions, so-called Lewy neurites, are found in neuronal processes. In MSA, aSyn inclusion pathology is predominantly present in oligodendrocytes. Besides these macroscopic morphological differences, indirect pathological evidence for different aSyn aggregate structures in different α-synucleinopathies was reported[1,3,4]. First structural insights into aSyn aggregates were obtained by solid-state NMR and cryo-electron microscopy of aSyn fibrils produced in the test tube[7–9]. Molecular level insight into aSyn aggregate structure in different α-synucleinopathies is however lacking.

To gain insight into the molecular structure of protein aggregates in different α-synucleinopathies, we amplified aSyn aggregates from brain extracts of patients pathologically confirmed with PD and MSA using protein misfolding cyclic amplification (PMCA). Fibril amplification by PMCA resembles the process of seeded aggregation, which underlies spreading of aSyn aggregates from cell to cell[10,11]. The brain-extract amplified aSyn fibrils were structurally characterized at the molecular level using hydrogen–deuterium (HD) exchange coupled to nuclear magnetic resonance (NMR) spectroscopy, electron paramagnetic resonance (EPR) and the specific binding of fluorescent probes. In addition, we analyzed several in vitro aSyn polymorphs using the same methods.

## Results

**Amplification of aSyn aggregates from brain extracts.** PMCA has been previously demonstrated to amplify aggregates of different proteins from in vivo aggregates[10], as well as from blood[12] and cerebrospinal fluid[13]. In addition, PMCA was used for aSyn anti-aggregating drug screening[14]. We used PMCA to amplify aSyn aggregates from brain extracts of ten patients pathologically confirmed with either PD or MSA (Table 1). Western blot analysis with an aSyn-specific antibody, as well as fluorescence measurements using the amyloid-binding dye thioflavin-T (ThT) showed that we successfully amplified aSyn aggregates from the

brain extracts (Fig. 1a; Supplementary Fig. 1)[15]. Control PMCA experiments with brain extract from an individual, in which an α-synucleinopathy was excluded, did not amplify aSyn aggregates (Fig. 1a; Supplementary Fig. 1).

To obtain sufficient quantities for structural analysis, PMCA-amplified amyloid fibrils were used in a second step to seed recombinant aSyn (Fig. 1b; Supplementary Fig. 2). Only in the presence of seeds did aSyn aggregate into ThT-positive, β-sheet rich amyloid fibrils (Figs. 1b, c, 2).

**Preparation of in vitro polymorphs.** In addition, we prepared different aSyn fibrils not specific for individual patients (termed in vitro polymorphs). One in vitro polymorph was generated by adding brain extracts from different patients to the PMCA reaction, but doubling both the concentration of recombinant aSyn and the PMCA duration. Because of the higher aSyn concentration and the longer PMCA duration, de novo aggregation of aSyn occurs. The resulting aSyn fibrils serve as an important control, because they represent an in vitro polymorph, which was generated under exactly the same buffer conditions as the fibrils amplified from brain extracts.

Two more aSyn polymorphs were produced through aggregation of recombinant aSyn in the absence of PMCA-seeds under two different buffer conditions: 50 mM Tris-HCl, 150 mM KCl, pH 7.5, 0.02% NaN$_3$ (termed "de novo hsAsyn", i.e. high-salt), and 5 mM Tris-HCl, pH 7.5, 0.02% NaN$_3$ (termed "de novo lsAsyn", i.e. low-salt)[5,6]. Previous studies showed that de novo hsAsyn and de novo lsAsyn differ in their fibril structure and also in their toxicity when added to cells or injected into animal models of aSyn-disease[5,6]. In addition, de novo hsAsyn and de novo lsAsyn fibrils were used for seeding of aSyn aggregation, in order to investigate the influence of the seeding reaction on aSyn fibril structure. These fibrils were termed "hsAsyn" and "lsAsyn". Circular dichroism and electron microscopy showed that hsAsyn and lsAsyn are amyloid fibrils with different morphologies (Fig. 2).

Finally, aSyn was aggregated de novo in 50 mM HEPES, 100 mM NaCl, pH 7.4, 0.02% NaN$_3$, followed by seeded aggregation in the same condition. This provided yet another in vitro polymorph (termed "high salt [HEPES]"), which can be compared to aSyn fibrils amplified from brain extracts.

**Binding of fluorescent dyes to aSyn fibrils.** Fluorescent dyes that bind to specific amyloid conformations can differentiate prion strains and determine inter-subject variability of amyloid-β aggregates in Alzheimer's disease[16,17]. To gain insight into the conformational differences between brain-extract amplified fibrils and in vitro aSyn polymorphs, we performed fluorescence measurements in presence of the amyloid-binding dye curcumin (Fig. 3a)[17,18]. The fluorescence spectra of curcumin in the presence of aSyn fibrils amplified from PD and MSA brain extracts

**Table 1 Demographics of donor brains from α-synucleinopathy patients.**

| Disease type # | Age | Gender | Postmortem delay (hrs) | Cause of death | Disease duration (yrs) |
|---|---|---|---|---|---|
| PD1 | 79 | Male | 17 | Acute myocardial infarction | 7 |
| PD2 | 82 | Female | 9 | Pneumonia | 8 |
| PD3 | 73 | Male | 20 | Cardiorespiratory failure | 13 |
| PD4 | 79 | Male | 42 | Septicemia | 17 |
| PD5 | 84 | Male | 5 | Cholangitis, cholangiocarcinoma | 12 |
| MSA1 | 82 | Male | 8 | Cardiorespiratory failure | 7 |
| MSA2 | 71 | Female | 19 | Hypostatic pneumonia | 6 |
| MSA3 | 61 | Male | 7 | Cardiorespiratory failure | 2 |
| MSA4 | 74 | Female | 18 | Sepsis | 7 |
| MSA5 | 69 | Male | 41 | Aspiration pneumonia | 8 |

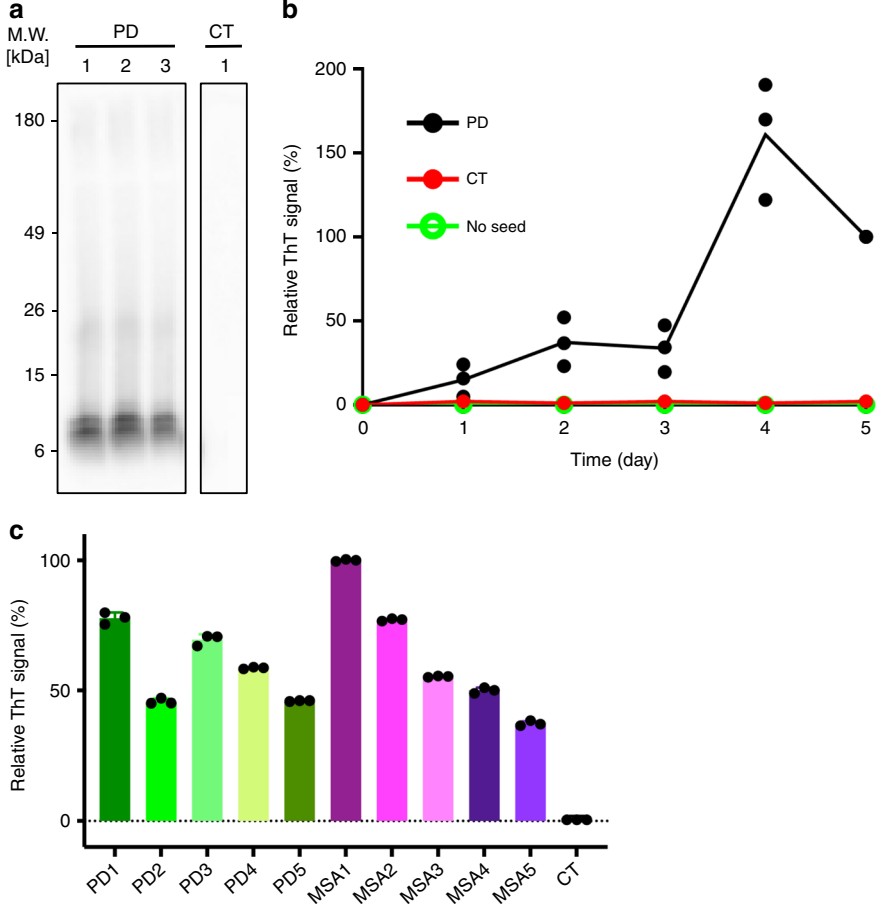

**Fig. 1 Amplification of aSyn aggregates from brain extracts. a** Immunoblotting of PMCA products with proteinase-K (PK) digestion. Similar PK-resistant signals were detected in PD samples (PD1–PD3), whereas nothing remained in the control sample (CT; brain extract from an individual, in which an α-synucleinopathy was excluded). An uncropped image of the blot is shown in Supplementary Fig. 1. **b** ThT-binding kinetics of recombinant aSyn with or without PMCA products as seeds. ThT-intensities for three PD samples (PD1–PD3) were averaged and used to derive error bars. **c** Relative ThT-fluorescence after seeding with different PMCA products. Error bars indicate standard deviation over three fluorescence measurements for each sample.

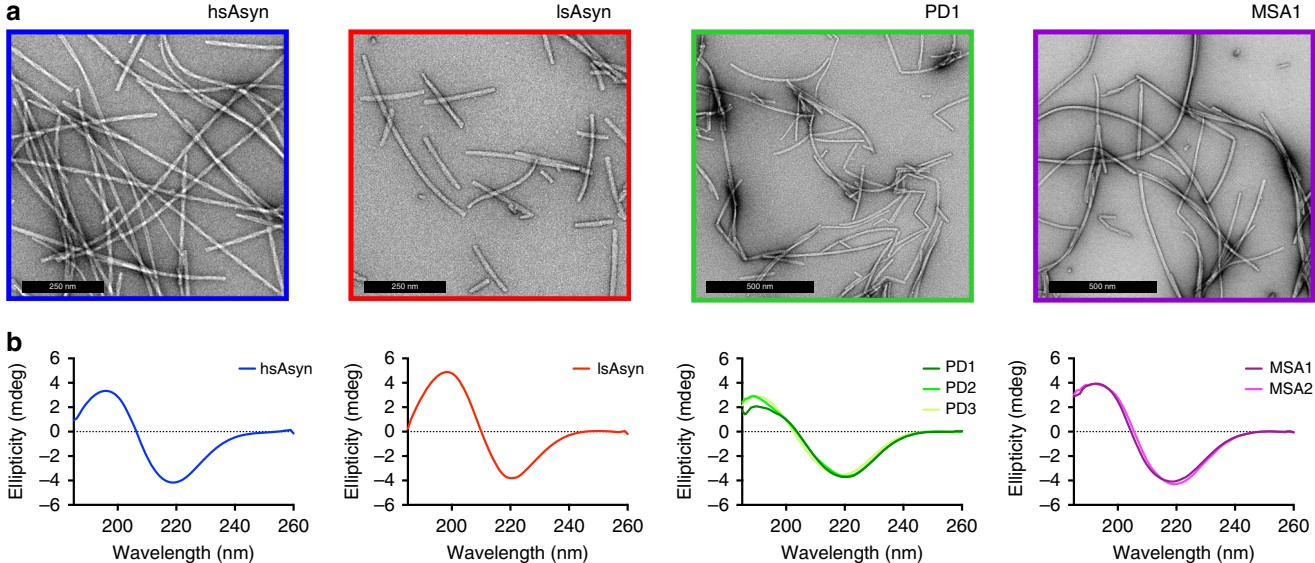

**Fig. 2 Electron micrographs and circular dichroism spectra of aSyn fibrils. a** Electron micrographs of the two in vitro polymorphs hsAsyn (blue) and lsAsyn (red), as well aSyn fibrils amplified from brain extracts of a PD (green) and a MSA patient (purple; Table 1). **b** Circular dichroism spectra of aSyn fibrils.

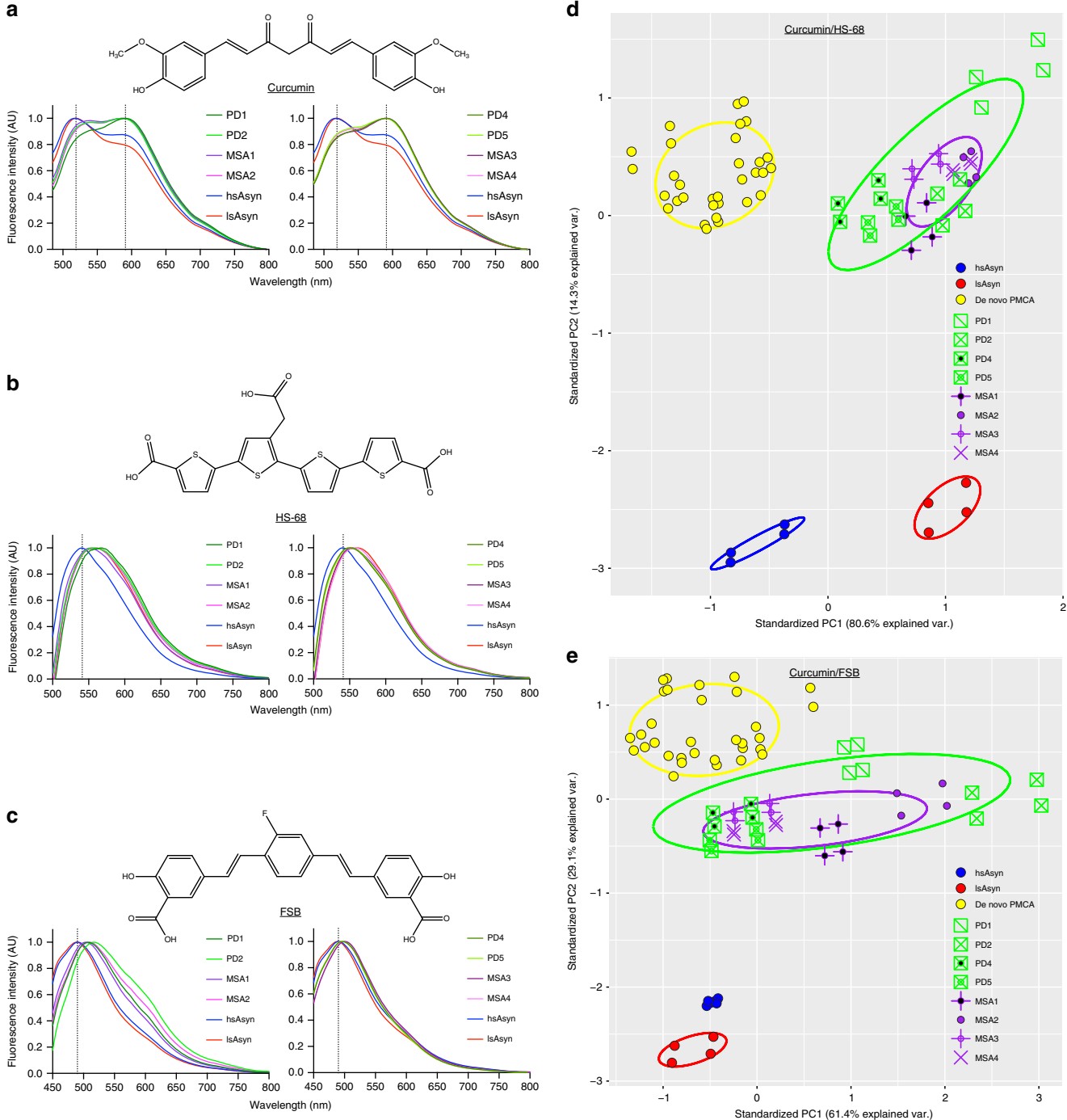

**Fig. 3 Fluorescent dyes distinguish aSyn aggregate structures. a–c** Normalized fluorescence spectra of curcumin (**a**), HS-68 (**b**) and FSB (**c**) in the presence of amyloid fibrils amplified from brain extracts of different patients (Table 1; PD/green, MSA/purple), as well as two in vitro aSyn polymorphs (hsAsyn/blue, lsAsyn/red)[5,6]. **d, e** PCA of fluorescence spectra of the amyloid-binding dyes curcumin and HS-68 (**d**), and curcumin and FSB (**e**), in presence of aSyn fibrils. aSyn fibrils amplified from patient brain-extracts are identified according to Table 1. In addition to brain-extract amplified fibrils, the fluorescence spectra of various in vitro polymorphs were analyzed: hsAsyn (blue), lsAsyn (red), and amyloid fibrils formed through de novo aggregation under conditions of PMCA (termed de novo PMCA; yellow). For each sample, fluorescence spectra of the three dyes were measured independently twice, and the resulting values were joined combinatorially resulting in four data points per sample in order to represent the intrinsic variability in the fluorescence measurements.

peaked at ~593 nm (Fig. 3a). In addition, a second local maximum was present at ~519 nm. In the case of in vitro polymorphs hsAsyn and lsAsyn, the relative intensity of the two maxima was inverted and the highest fluorescence intensity was observed at ~519 nm (Fig. 3a).

As second dye, we used HS-68 (Fig. 3b; Supplementary Fig. 3). HS-68 was previously applied to brain-tissue sections from

transgenic mice and the variation in the emission profiles from HS-68 were used to distinguish polymorphic Aβ and tau deposits[18,19]. The fluorescence spectra of HS-68 are characterized by a single fluorescence maximum (Fig. 3b). This maximum was shifted from ~541 to ~561 nm when comparing hsAsyn and lsAsyn, respectively (Fig. 3b). The position of the fluorescence maxima of the fibrils amplified from PD and MSA brain extracts

were more similar to that of lsAsyn, i.e. from ~551 to ~567 nm. Fibrils from PD patient 1 were most red-shifted among the brain-amplified fibrils.

The congo red derivative (E,E)-1-fluoro-2,5-bis(3-hydroxycar-bonyl-4-hydroxy) styrylbenzene (FSB; Fig. 3c) stains Aβ plaques in the brain[20]. Upon binding to amyloid fibrils the fluorescence intensity of FSB increases[20], with different types of aSyn fibrils peaking at different wavelengths (Fig. 3c). In contrast to HS-68, the normalized fluorescence spectra of FSB were very similar for the two in vitro polymorphs hsAsyn and lsAsyn, while FSB spectra of aSyn fibrils amplified from brain extracts were red-shifted. In addition, the shape of the FSB spectra diverged between aSyn fibrils amplified from brain extracts of different patients (Fig. 3c).

**Fluorescent dyes distinguish aSyn aggregate structures**. The above data show that both the profile and the peak wavelength of the fluorescence spectra of curcumin, HS-68 and FSB change for different aSyn fibrils. To fully take advantage of this sensitivity and increase the discriminative power, we performed principal component analysis (PCA). To this end, the normalized fluor-escence spectra of two dyes for a specific aSyn fibril type were concatenated and treated as a single spectrum. Because curcumin fluorescence spectra showed the largest diversity, we combined the buffer baseline-corrected and normalized fluorescence spectra of curcumin with either HS-68 or FSB. The data points from these artificial spectra (curcumin + HS-68: 157 + 163 data points; curcumin + FSB: 157 + 166 data points) of the different aSyn fibrils were subsequently analyzed by PCA (Fig. 3d, e). The two largest principal components (PCs) accounted for more than 90% of the variations in the spectra of curcumin, HS-68 and FSB.

The two widely studied in vitro polymorphs hsAsyn and lsAsyn clustered at different locations in the two-dimensional PCA plots (Fig. 3d, e), in agreement with conformational

differences established by solid-state NMR spectroscopy[9]. The de novo PMCA fibrils (yellow data points in Fig. 3d, e), i.e. the fibrils that were obtained through de novo aggregation in the presence of brain extracts (PD1, PD2, PD4, PD5, MSA1, MSA2, MSA3, and MSA4 according to Table 1), also formed a single cluster, which did not overlap with either the hsAsyn or the lsAsyn cluster (Fig. 3d, e). The analysis showed that aggregation of recombinant aSyn under different in vitro conditions generates amyloid fibrils with distinct conformational properties.

The spectral properties of brain-extract amplified aSyn fibrils did not overlap with the in vitro polymorphs hsAsyn and lsAsyn, but formed a distinctive spectral cluster (Fig. 3d, e). In addition, the cluster of the brain-extract amplified fibrils did not overlap with the cluster of the de novo PMCA fibrils, i.e. those fibrils which were formed through de novo aggregation in the presence of brain extracts in the PMCA assay—that is under identical conditions except for the doubling of the recombinant aSyn concentration and PMCA duration. The analysis further showed that considering all PD and MSA patients, the three dyes did not distinguish between the two different α-synucleinopathies (Fig. 3d, e). Notably, however, the spectral properties of the MSA-amplified aSyn fibrils were quite homogeneous and resulted in focal clusters in the two PCA plots. In contrast, PD-amplified fibrils formed a more widespread cluster, suggesting a greater conformational diversity (Fig. 3d, e). The extent of variability of the PD-amplified fibrils was also greater than that of the de novo PMCA fibrils.

**Single-residue analysis of aSyn fibril structure**. To gain insight into the structure of aSyn aggregates at the molecular level, we performed NMR-based HD exchange (Fig. 4). When HD exchange is coupled to rapid fibril dissociation, deuterium incorporation levels can be determined with single-residue reso-lution by NMR[21,22]. Previous HD exchange studies conducted on

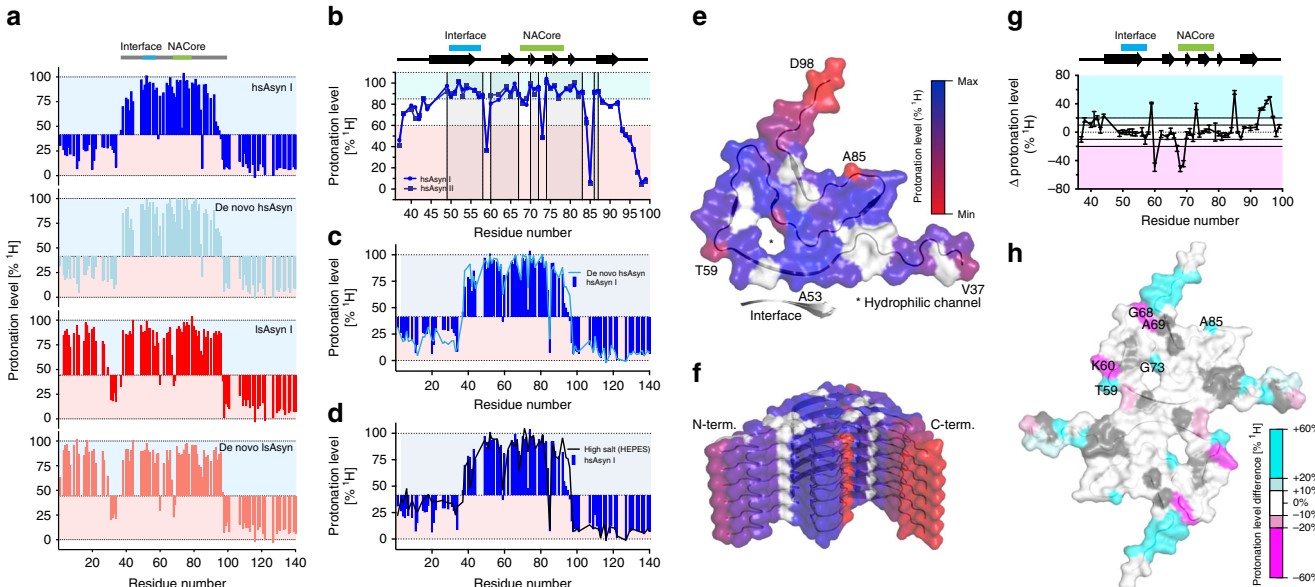

**Fig. 4 H/D exchange properties of aSyn in vitro polymorphs. a–d** Residue-specific HD-exchange profiles of different in vitro polymorphs (see text for further details). **e** Mapping of residue-specific protonation levels of hsAsyn on the structure of the hsAsyn filament (PDB code: 6A6B). The interface to the second filament, which involves A53, is marked. The scale bar indicates protonation levels relative to the minimum protonation level observed for V37-Q99. Residues with signal overlap were excluded from the analysis (shown in white). **f** Side view of (**e**). **g**, **h** Residue-specific differences in the protonation levels of hsAsyn and lsAsyn in the region from V37 to Q99 (**g**), mapped in (**h**) onto the core structure of hsAsyn fibrils (PDB code: 6A6B). Difference values were calculated on the basis of the average protonation levels observed for two independently seeded samples (hsAsyn I/II and lsAsyn I/II, respectively). Error bars represent std. NACore is the most hydrophobic part of the aSyn sequence. The interface between two aSyn filaments as seen in the structure of hsAsyn (PDB code: 6A6B) is marked.

in vitro aggregated aSyn fibrils established that a forward exchange time of 1.5 days allows robust determination of residue-specific protonation levels[21,23]. To further enhance the distinction for residues with different amide proton exchange rates, we used a mixture of $H_2O$ and $D_2O$ in the dissociation buffer (Supplementary Fig. 4)[21].

Firstly, we determined the HD exchange profile for the hsAsyn polymorph (Fig. 4a, blue). In agreement with previous data[21,23], we found that the fastest solvent exchange occurs in residues located in the ~40 C-terminal residues for which less than 25% of amide protons remained after 1.5 days of forward exchange in $D_2O$ (Fig. 4a). Extensive incorporation of deuterium was also observed for the first 36 residues at the N-terminus of aSyn (Fig. 4a). In contrast, residues L38-V92 retained a large fraction of their amide protons after three days of exchange in $D_2O$ (Fig. 4a), indicating that these amide protons participate in hydrogen bonds.

The hydrogen-bonded core of hsAsyn fibrils as determined by HD was consistent with that defined by solid-state NMR and cryo-electron microscopy for wild-type aSyn and aSyn(1–121)[7–9]. Fast solvent exchange was undertaken by V37-T44 and G93-Q99, which are located in the N- and C-terminal parts, respectively, and are not surrounded by additional cross-β-sheet structure in hsAsyn (Fig. 4e, f). A connection between solvent accessibility and differences in protonation levels was further supported by the finding that residues next to the hydrophilic channel[7–9,24] displayed decreased protonation levels (Fig. 4e). In addition, faster solvent exchange was observed for some turn residues (e.g. T59, A85; Fig. 4e, f), potentially due to lower hydrogen bond strength[25]. In contrast, we detected high protonation levels for the residues A53-A56, which form the interface with a second filament in hsAsyn fibrils (Fig. 4a, e)[7,8].

hsAsyn fibrils (termed hsAsyn fibrils II), which were generated through independent aggregation of recombinant aSyn in presence of the same seeds as hsAsyn (termed hsAsyn I in Fig. 4a–d), displayed highly similar protonation levels in the cross-β structure region (Fig. 4b). In addition, the in vitro polymorph "high salt [HEPES]" displayed a similar HD exchange

profile (Fig. 4d). Notably, HD exchange profiles of de novo aggregated and in vitro seed-derived hsAsyn fibrils were highly similar (Fig. 4a, c), indicating the propagation of amyloid fibril structure during seeding.

**HD exchange profile of lsAsyn**. We then determined the HD exchange profile of the lsAsyn polymorph (Fig. 4a; red). In agreement with solid-state NMR spectroscopy[5], the lsAsyn polymorph contains hydrogen-bonded structure in the ~30 N-terminal residues. Comparison of the HD exchange profiles of de novo and seeded lsAsyn underscored the propagation of amyloid structure during seeding (Fig. 4a). In addition, differences in the protonation levels in the central part were evident when comparing lsAsyn and hsAsyn (Fig. 4g). In the lsAsyn polymorph, solvent exchange for residues V37-T44 and G93-K97 was decreased when compared to hsAsyn (Fig. 4g). The turn residues T59, K60, G68, A69, G73, and A85 also differed in the protonation levels (Fig. 4g, h). Thus, changes in amyloid structure occur when aSyn is aggregated at a low ionic strength[5].

**Molecular insights into fibrils amplified from brain extract**. To gain molecular level insight into the structure of patient-associated aSyn aggregates, we performed HD exchange studies of the aSyn fibrils amplified from the brain extracts of the five PD and five MSA patients (Fig. 5a; left and middle column). To unambiguously distinguish de novo aggregation from seed-dependent amplification, we also determined the HD exchange profiles of six de novo PMCA fibril samples (four of them shown in the right column of Fig. 5a). The de novo PMCA HD exchange profiles were all very similar with only small variations in residue-specific protonation levels (Fig. 5a; right column, bottom). Independent of whether the brain extract was from a PD or a MSA patient, de novo aggregation thus overrides the conformational signature of the seed. In addition, the conformational properties of the de novo PMCA fibrils were similar but not identical to those of the in vitro polymorph or de novo hsAsyn, in

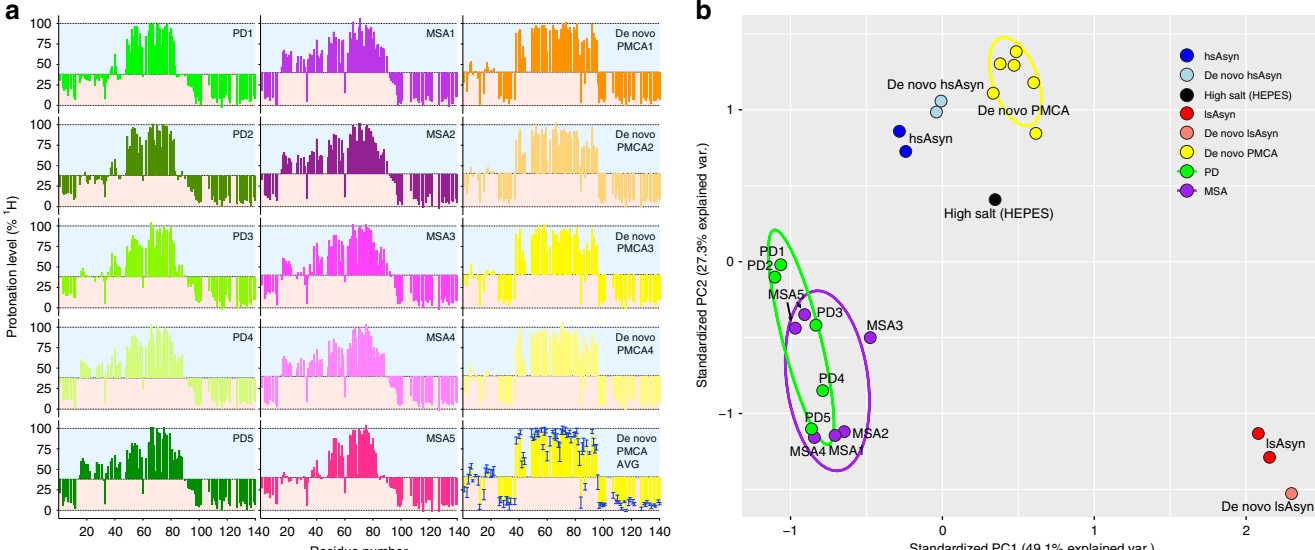

**Fig. 5 Molecular level insights into the structure of aSyn fibrils amplified from patient brain extracts. a** HD-exchange profiles of aSyn fibrils amplified from brain extracts of PD patients (left, green) and MSA patients (middle, purple). HD-exchange profiles of four de novo PMCA aSyn fibrils, which are not based on seeding/propagation, are shown in the right column (yellow to orange) together with the residue-specific average values for six de novo PMCA fibril samples. Error bars represent std. **b** PCA of protonation levels in aSyn fibrils (see text for further details). Independent repetition of HD exchange measurements for fibrils obtained from patient MSA5 are included (indicated by black arrows).

agreement with similar (in particular similar ionic strength) but not identical aggregation conditions.

The HD exchange profiles of fibrils amplified from the ten different patients diverged substantially from all in vitro polymorphs (Figs. 4, 5a). In the case of samples PD1 and PD2, protonation levels of residues L38-T44 were lower when compared to those of fibrils not seeded by brain extracts (Figs. 4, 5a). In addition, the protonation levels decreased significantly from E83, suggesting a loss of hydrogen bonds (or decrease in hydrogen bond strength) for E83-K96 in PD1/PD2-derived fibrils (Fig. 5a). Thus, amyloid fibrils, which were amplified from PD patients 1 and 2, have a smaller hydrogen-bonded core than fibrils that were not seeded with brain extracts (Figs. 4, 5a). This smaller core contains the most hydrophobic part of the aSyn sequence, the NACore region (G68-A78), which can form amyloid fibrils in isolation[26].

The NAcore also displayed the highest protonation levels in the fibrils amplified from the other three PD patients (Fig. 5a). However, protonation levels of residues A53-A56, which form the interface with a second filament in hsAsyn fibrils[7,8], were less protected from solvent exchange in PD3, PD4, and PD5 fibrils. At the same time, PD4 and PD5 fibrils had higher protonation levels than PD1, PD2, and PD3 fibrils for the residues A19-A29 (Fig. 5a). The HD exchange profiles of aSyn fibrils amplified from brain extracts of different PD patients are thus more heterogeneous than those of the in vitro polymorphs, including the de novo PMCA fibrils (Fig. 5a).

The HD exchange profiles of the fibrils amplified from the five MSA patients are shown in the middle row of Fig. 5a. They appear less heterogeneous when compared to the PD profiles. However, the MSA5 profile resembles more that of PD1 and PD2 than of MSA1 to MSA4.

**Classification of HD exchange profiles of aSyn aggregates.** In Figs. 4 and 5, H/D protonation levels of 99 non-overlapping residues in more than 20 aSyn fibril samples are shown. To allow an unbiased comparison of all protonation levels, this high-dimensional data set was transformed using PCA (Fig. 5b). The two largest principal components accounted for 76.4% of the variations in the HD data and clustered the de novo PMCA fibrils in a small region of the PCA plot (Fig. 5b). Closest to this cluster are the hsAsyn fibrils (both de novo and seeded). In addition, the largest principal component of the high salt [HEPES]" fibrils was similar. In contrast, both de novo aggregated and seeded lsAsyn fibrils clustered to very different regions of the PCA plot (Fig. 5b). The distinct PCA classification of lsAsyn is in agreement with the observation that only lsAsyn but none of the other aSyn fibrils studied in this work contain hydrogen-bonded structure within the N-terminal 20 residues (Figs. 4, 5).

PCA distinguished the brain-extract amplified fibrils from all in vitro polymorphs (Fig. 5b). All brain-extract amplified fibrils had a similar PC1 component, but showed some spread in PC2. Similar to the PCA analysis of the fluorescence spectra of the three dyes, the aSyn fibrils amplified from MSA brains clustered in a narrower region than those of PD brains. In agreement with the observed residue-specific differences in the protonation levels (Fig. 5a), PD1 and PD2 were most distinct from MSA1 and MSA2 (Fig. 5b).

To gain insight into the conformational basis of this distinction, we compared the HD profile averaged over PD1 and PD2 with that averaged over MSA1 and MSA2 (Fig. 6). The comparison showed that V37-T44 and A85-G93 in MSA1/MSA2-amplified aSyn fibrils exchanged their amide protons less rapidly when compared to PD1/PD2 fibrils (Fig. 6a, b). In addition, the protonation levels of A53-A56 of MSA1/MSA2-amplified fibrils

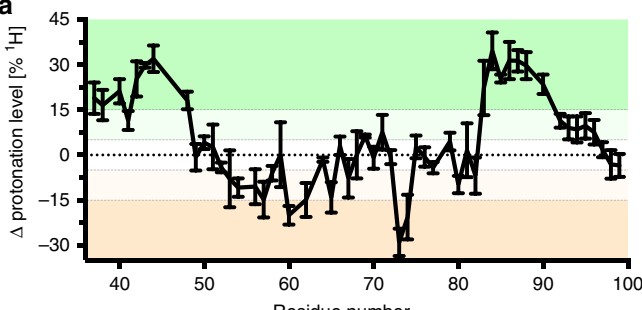

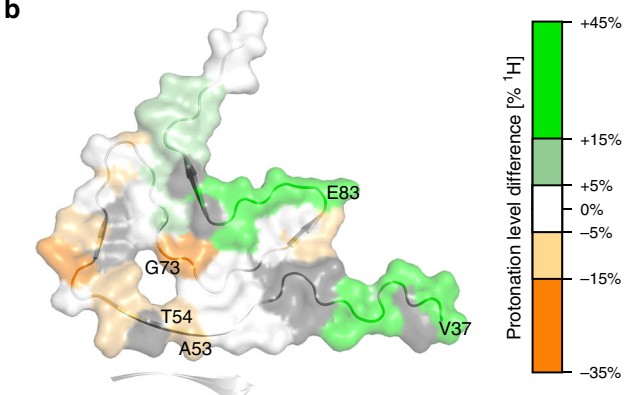

**Fig. 6 Structural diversity in aSyn aggregates amplified from patient brain extracts. a, b** Differences between protonation levels in aSyn fibrils amplified from PD1/PD2- and MSA1/2-brain extracts, mapped in (**b**) on the core structure of hsAsyn fibrils. Difference values were calculated on the basis of the average protonation levels observed for PD1 and PD2 (and MSA1 and MSA2, respectively). Errors were calculated from the differences in protonation levels between the two patient profiles of either PD or MSA. Residues that experience faster solvent exchange in MSA1/2 fibrils (when compared to PD1/PD2 fibrils) are shown in orange, those with slower solvent exchange in green. Residues with signal overlap were excluded from the analysis (shown in gray in (**b**)).

were 5–15% lower than those in PD1/PD2-amplified fibrils (Fig. 6a, b). It is known that A53-A56 form the interface with a second filament in hsAsyn fibrils[7,8]. This suggests that the aSyn fibrils amplified from MSA patients 1 and 2 differ from those amplified from PD patients 1 and 2 in both tertiary and quaternary structure.

**EPR spectroscopy of aSyn fibrils.** To further analyze the molecular properties of aSyn fibrils, we used EPR. For this analysis, recombinant aSyn-T54C/A90C mutant protein (Supplementary Table 1) was labeled with the nitroxide tag MTSL at the two cysteine residues (Fig. 7a), and mixed with wild-type aSyn at a molar ratio of 1:30 (aSyn-T54C/A90C:aSyn)[27]. This mixture was subsequently aggregated into amyloid fibrils through seeding with brain-extract amplified fibrils (or in vitro aSyn polymorphs as control). Continuous wave EPR spectra of all samples showed broad EPR lines (Fig. 7b, c), indicative of the incorporation of MTSL-labeled aSyn-T54C/A90C protein into fibrils.

Pulsed EPR measurements revealed the distance between the two MTSL labels to be ~2.4–2.6 nm (Fig. 7d, e)[27]. This distance is in agreement with EPR data on multiple MTSL-labeled aSyn proteins[27] and the structure of hsAsyn (Supplementary Fig. 5), demonstrating that addition of MTSL-labeled mutant protein to the seeding assay at high dilution (1:30 aSyn-mutant:aSyn) did

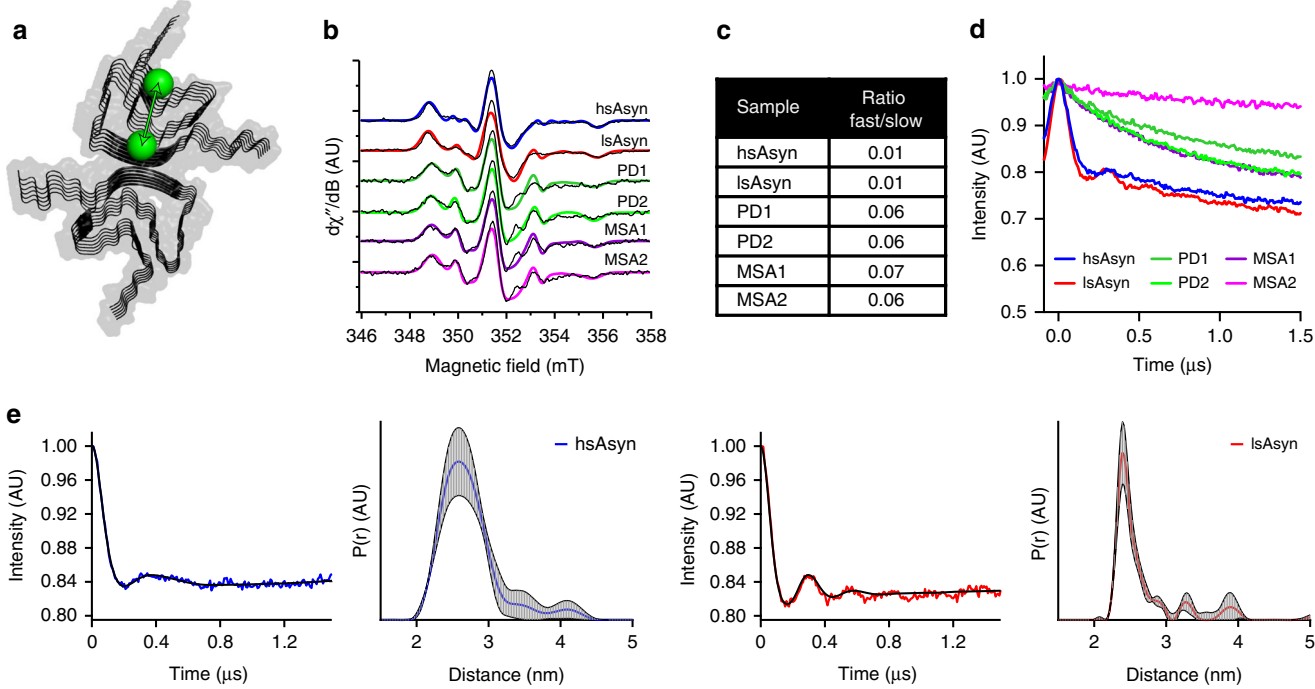

**Fig. 7 EPR spectroscopy of aSyn fibrils. a** Schematic representation of the location of MTSL spin labels (green balls) attached to residues 54 and 90. **b** Continuous wave EPR spectra of aSyn fibrils. Fits derived from a two-spin simulation are shown as colored lines. **c** Dynamic parameters (weight: fast/slow) of MTSL spins derived from fitting continuous wave EPR spectra. For both spin species, $g \approx$ [2.008, 2.006, 2.002] and $A/h \approx$ [17.8–19 18.5–19 93–110] MHz. The peak-to-peak isotropic linewidths (lwpp) for the two species were lwpp$_{(slow)}$ = 0.52–0.67 mT and lwpp$_{(fast)}$ = 0.3–0.4 mT, respectively. **d** Dipolar modulation (without background correction) of aSyn fibrils from 34 GHz four-pulse PELDOR experiments. **e** Dipolar modulation (corrected for background) and normalized distance distribution determined by 34 GHz four-pulse PELDOR experiments for hsAsyn (blue) and lsAsyn (red).

not perturb aSyn fibril structure. A similar yet slightly shorter distance was determined for lsAsyn (Fig. 7e), suggesting structural differences between lsAsyn and hsAsyn.

In contrast to the pulsed EPR data for the in vitro aSyn polymorphs, no dipolar modulation was observed in any of the aSyn fibrils amplified from brain extracts, despite identical experimental parameters and comparable signal-to-noise ratios (Fig. 7d). The lack of dipolar modulations for the T54C/A90C-interaction in aSyn fibrils amplified from brain extracts suggests that these two residues do not have a single defined distance in these fibrils. This could be caused by a number of reasons (alone or combined): higher flexibility of the MTSL side chains, increased flexibility of the backbone, and larger heterogeneity/polymorphism in aSyn fibril conformations. In addition, estimation of MTSL mobility on the basis of continuous wave EPR spectra suggested that at least one of the two MTSL labels was more flexible in aSyn fibrils amplified from brain extract when compared to aSyn in vitro polymorphs (Fig. 7c). Because HD-exchange studies showed that the cross-β-sheet structure of residues A85-G93 is destabilized in fibrils amplified from patient brain extracts (Fig. 5a), the MTSL with higher flexibility could be located at position 90.

## Discussion

Different aSyn aggregate structures have been suggested as possible sources of the clinical differences between PD and MSA[1,3,4]. In addition, different types of aSyn fibrils produced in the test tube cause different degrees of neuronal toxicity[5,6]. In order to gain molecular level insight into these properties, we amplified aSyn aggregates from brain extracts of patients pathologically confirmed with PD or MSA using PMCA (Figs. 1, 2). Amplification of aSyn fibrils from patient brain extracts mimics the process of seed-dependent misfolding and aggregation of proteins

in host brain cells. In addition, we generated several in vitro aSyn polymorphs (Fig. 2).

The conformational properties of the aSyn fibrils were analyzed using fluorescent probes (Fig. 3), NMR spectroscopy (Figs. 4–6) and EPR (Fig. 7). The strength of the spectroscopic approaches is that all molecules contribute to the measurement. They thus characterize the full spectrum of fibril conformations present in each sample. This is important, because amyloid fibrils are often polymorphic[28–30]. The chosen spectroscopic approach is therefore complementary to cryo-electron microscopy of amyloid fibrils purified from patient brain[31].

A major finding of our study is that aSyn fibrils amplified from patient brain extracts were structurally different to all of the in vitro polymorphs analyzed (Figs. 3–5, 7). Previous cryo-electron microscopy studies have shown that the structure of in vivo tau aggregates differs from tau fibrils aggregated in vitro[31–34]. Although these findings are based on relatively small numbers of patients, they highlight the potential differences between in vitro and in vivo aggregates. Potential reasons for these differences are currently unknown, but could involve post-translational modifications or the presence of co-factors. Consistent with the latter hypothesis, additional/unresolved electron density in the three-dimensional structure of tau fibrils from chronic traumatic encephalopathy was attributed to an unknown hydrophobic molecule[33]. The observed conformational differences between amyloid fibrils propagated from recombinant aSyn and brain-derived aSyn fibrils raise questions regarding the suitability or specificity of using the former in cell and animal studies of PD and MSA.

Another important observation is that the structural diversity was larger in the case of aSyn fibrils amplified from brain extracts of the five PD patients when compared to MSA-propagated fibrils (Figs. 3d, e, 5b). Indeed, PD exhibits a wide variability of disease

phenotypes and progression, in some cases demonstrating a prodromal period lasting decades and a median post-diagnosis survival in excess of a decade[2]. In contrast, MSA is consistently more aggressive, with earlier age of onset and a median survival from date of onset of ~9 years (Table 1)[2]. The structural variability/heterogeneity of aSyn aggregates in neurodegenerative diseases is supported by data from proteinase-K digestion of aSyn fibrils isolated from pathologically confirmed MSA brain, showing three major bands[35]. The range in aSyn aggregate structures within a single disease phenotype is also in agreement with the polymorphic morphology of aSyn fibrils isolated from PD patient brains[29], and the variations in amyloid-β fibril structure amplified from brain extracts of patients with Alzheimer's disease[30]. Although larger patient numbers are required to link aSyn aggregate structure with symptomatic heterogeneity of the disease, our data point to the possibility that aSyn aggregate structure could be specific for individual patients or certain subtypes of PD.

One contentious finding from our study is that aSyn fibrils propagated from PD and MSA patients do not exhibit distinctive structural properties (Figs. 3, 5). This is inconsistent with a *one disease-one strain* hypothesis, i.e. a unique connection between the clinical disease presentation and a single, defined structure of aSyn fibrils. Currently, the connection between aSyn disease type and aSyn structure is based on a number of indirect observations, including differences in macroscopic morphology of aSyn inclusions in patients[2], differences in seeding potential/kinetics of material from different α-synucleinopathies[4,36], differences in antibody binding to different aSyn aggregates (e.g. [3]), the ability to produce in vitro different aSyn fibril structures[5,6]. These observations could, however, also be caused by other factors besides differences in aSyn aggregate structure, such as differences in cellular environment, e.g. in PD, aSyn aggregates occur predominantly in neurons, whereas in MSA they are found in oligodendrocytes[2], differences in genetic backgrounds in patients, different burden of aSyn aggregates in different diseases/patient samples, as well as differences in post-translational modifications of aSyn aggregates. The macroscopic morphology of aSyn aggregates in patients is also on a different length scale than the molecular structure of amyloid fibrils. In addition, aSyn-rich aggregates in patients not only contain aSyn fibrils but many other components[37], which potentially influence both the morphology of insoluble deposits and their potential for seeding pathogenic aggregation. Another aspect that might require attention is the region of the brain from which aSyn aggregates are extracted. In the current study, amygdala was selected. Thus, it could be possible that different aSyn aggregate structures exist in different regions of the brain.

## Methods

**Materials.** Isopropyl-1-thio-β-D-galactopyranoside (IPTG) and protease inhibitor cocktail were purchased from Sigma-Aldrich (St. Louis, MO, USA). Anion-exchange chromatography (Mono Q, 5/50 GL) and size-exclusion (Superdex 200, 26/600) columns were purchased from GE healthcare (Fairfield, Connecticut, USA). Teflon beads with a diameter of 2.38 mm were purchased from SmallParts. FSB [(E,E)-1-fluoro-2,5-bis(3-hydroxycarbonyl-4-hydroxy)styrylbenzene] and curcumin were purchased from Santa Cruz Biotechnology and Sigma-Aldrich, respectively. HS-68 was synthesized according to published procedures[19].

**Preparation of brain extracts.** Ethical approval to access and work on the human brain tissues was given by the Human Research Ethics Committee of the University of New South Wales. Following approvals, brain tissues were received from the Sydney Brain Bank at Neuroscience Research Australia which is supported by The University of New South Wales and Neuroscience Research Australia. Human amygdalas were sonicated with Vibra-cells (Sonics, Newtown, CT, USA) to 10% weight/volume (w/v) solution with homogenizing buffer (1% Triton X-100, Protease Inhibitor Cocktail in PBS). Sonicated samples were centrifuged at 3000 g for 40 s. Protein concentrations in supernatants were determined by the bicinchoninic acid assay (Pierce, Rockford, IL, USA).

**Recombinant aSyn preparation.** N-terminally acetylated aSyn was obtained by co-transfection of *E. coli* BL21 (DE3) cells with pT7–7 plasmid encoding for human aSyn (kindly provided by the Lansbury Laboratory, Harvard Medical School, Cambridge, MA) and *S. pombe* NatB acetylase complex[38] using pNatB plasmid (pACYCduet-naa20-naa25, Addgene, #53613, kindly provided by Dan Mulvihill). The mutant protein aSyn-T54CA90C was constructed using the QuikChange site-directed mutagenesis kit (Stratagene), and the introduced modifications were verified by DNA sequencing.

For aSyn expression and purification, transformed BL21 (DE3) cells were grown at 37 °C in LB medium to an $OD_{600}$ of 0.8 and shifted to 25 °C adding 0.5 mM IPTG for protein expression o/n. Cells were harvested by centrifugation on a Beckman Coulter Avanti JXN-26 centrifuge with a JLA-8.1 rotor at 12,000 g for 15 min at 4 °C. The obtained cell pellet was lysed by French press (Avestin EmulsiFlex-C3) in 20 mL lysis buffer (10 mM Tris-HCl, pH 8, 1 mM EDTA, 1 mM PMSF) per 1 L cell culture. The cell lysate was heated up to 96 °C in a water bath and incubated at this temperature for 30 min. The supernatant was collected by centrifugation (Beckman Coulter, JA-25.5 rotor, 22,000 g) at 4 °C for 30 min. Streptomycin sulfate was added to the supernatant at a final concentration of 10 mg mL$^{-1}$ and incubated at 4 °C for 15 min. The supernatant was collected by centrifugation (JA-25.5, 30 min, 22,000 g) and ammonium sulfate was added to a final concentration of 360 mg mL$^{-1}$ followed by incubation at 4 °C for 15 min. After a final centrifugation step, the protein pellet was obtained and dialyzed against 25 mM Tris-HCl, pH 7.7 o/n. The dialysate was applied to an anion exchange column (GE Healthcare, Mono Q 5/50 GL) and eluted with 25 mM Tris-HCl, pH 7.7, 2.0 M NaCl using a salt gradient from 0 to 1 M. The aSyn fraction eluted at 300 mM NaCl. High aSyn purity and buffer exchange was achieved by a final size-exclusion run (GE Healthcare, Superdex 75 10/300 GL) with 50 mM HEPES, 100 mM NaCl, pH 7.4, 0.02% NaN$_3$ on a GE Healthcare Äkta pure system.

$^{15}$N-labeled wild-type aSyn was expressed in *Escherichia coli* grown in M9 minimal medium supplemented with $^{15}$NH$_4$Cl (Cambridge Isotope Laboratories, Cambridge, MA) and purified as stated for *E.coli* in LB medium. Pure aSyn in 50 mM HEPES, 100 mM NaCl, pH 7.4, 0.02% NaN$_3$ was filtered through sterile 0.22-μm filters and stored at −80 °C.

**Spin labeling of aSyn.** The reaction of the aSyn-T54C/A90C mutant protein with the nitroxide spin label MTSL (1-oxy-2,2,5,5-tetramethyl-d-pyrroline-3-methyl)-methanethiosulfonate (Toronto Research Chemicals, Toronto) was carried out as described[39]. Briefly, DTT was removed from the buffer before labeling by using size-exclusion chromatography (PD-10 columns, Amersham Pharmacia Biosciences), and the proteins were equilibrated in 20 mM Tris-HCl buffer, pH 7.5, and 100 mM NaCl. Free sulfhydryl groups were reacted with a 5-fold molar excess of the MTSL solubilized in acetone, at 4 °C for 16 h. Unreacted spin label was removed using PD-10 columns, and spin-labeled proteins were concentrated using Microcon YM-3 (molecular weight cutoff, 3,000) centrifugal filters (Amicon).

**Mass spectrometry.** Acetylation of aSyn was validated using an SQD2 mass spectrometer (Waters) with ACQUITY Arc-System (column: XSelect Peptide CSH C18; 4.6 mm × 100 mm, 130 Å, 2.5 μm; running buffer: A: water + 0,1% TFA, B: ACN + 0,1% TFA).

**PMCA.** Equipment for PMCA, which includes the microplate horn (#431MPX), a sound enclosure (#432MP) and a thermoelectric chiller (#4900), was purchased from Qsonica (Qsonica, Newtown, CT, USA). PMCA was carried out with recombinant aSyn and tissue homogenates. aSyn monomers were prepared at 2.5 μM concentration in the conversion buffer (1% Triton X-100 in PBS) and 50 μL were transferred into PCR tubes that contained 2.5 μg of brain homogenates. Three Teflon beads were placed in the PCR tubes before adding the mixture. For PMCA, the samples were subjected to 48 cycles of 20 s sonication (amplitude 1%) and 29 min 40 s incubation at 37 °C[15]. PMCA product was analyzed by Western blot using the aSyn-specific antibody from BD Transduction, USA (#610787) at a dilution of 1:1500.

**Preparation of aSyn fibrils.** aSyn aggregations were performed at 37 °C using 250 μM monomeric, N-terminally acetylated aSyn taken from the supernatant (SN) of freshly thawed aSyn on ice after ultracentrifugation (UC) for 1 h and contained 0.02% NaN$_3$. All UC steps were performed on a Beckman Coulter Optima MAX-XP using a TLA 100.3 rotor with a rotor speed of 165,000 g. aSyn concentrations were determined by UV absorption (5960 M$^{-1}$ cm$^{-1}$) at 280 nm using a Eppendorf BioSpectrometer. Fibrillization was monitored by ThT fluorescence.

In the case of de novo fibrillization of high (hsAsyn; 50 mM Tris-HCl, 150 mM KCl, pH 7.5, 0.02% NaN$_3$) and low salt (lsAsyn; 5 mM Tris-HCl, pH 7.5, 0.02% NaN$_3$) in vitro polymorphs, monomeric aSyn was dialysed against the respective aggregation buffer at 4 °C o/n, followed by fibrillization as described in[5]. Fibrils were harvested by UC for 2 h at 25 °C and the SN was separated from the pellet. aSyn concentrations in the SN were determined spectrophotometrically. De novo aggregated fibril pellets were resuspended in fresh aggregation buffer to a final concentration of 250 μM, water bath sonicated for 1 min (Bandelin Sonorex Digitec water bath), aliquoted and shock-frozen in liquid nitrogen for storage at −80 °C. Seeded aggregation of the in vitro polymorphs was performed in the same manner

as de novo aggregation, but adding 0.5% (w/w) of room-temperature thawed seeds to the aggregation reaction. If not stated otherwise, all data reported in the manuscript were obtained for in vitro polymorphs obtained through seeding.

De novo aggregation for another high salt (50 mM HEPES, 100 mM NaCl, pH 7.4, 0.02% NaN$_3$) aSyn polymorph was achieved by agitation with magnetic stir bars at 300 rpm[40], followed by seeding as described above for hsAsyn/lsAsyn.

For the patient-derived aSyn fibrils, 0.5% (w/w) PMCA product was added to 250 µM aSyn stock solution (50 mM HEPES, 100 mM NaCl, pH 7.4, 0.02% NaN$_3$) and initially water bath sonicated for 10 min. This mixture was aggregated under quiescent conditions in 1.5 mL Eppendorf cups in a ThermoScientific Heratherm incubator.

**ThT fluorescence assay.** The assembly of aSyn fibrils was monitored over time by Thioflavin T (ThT) fluorescence. For each aggregation assay a fresh 10 µM ThT glycine buffer solution (50 mM glycine in deionized water, pH adjusted to 8.8 and sterile filtered) was produced from a 5 mM ThT stock in deionized water. 2 µl aliquots of aSyn aggregation sample were taken at different time points from the total sample volume and added to 200 µl of 10 µM ThT in glycine buffer to monitor the fibril assembly progress via a Cary Eclipse fluorescence spectrophotometer (Agilent Technologies). The samples were thoroughly mixed at room temperature and 180 µl were transferred to a black 96-well plate (Greiner) and the maximum emission value was determined at 482 nm exciting the sample well at 440 nm (ex. bandwith: 10 nm, em. bandwith: 10 nm, gain: 600 V PMT detector).

**Transmission electron microscopy.** Samples were adsorbed onto 400 mesh carbon-coated copper grids and the buffer was removed using a filter paper. Subsequently, samples were stained by the addition of 1% uranyl acetate solution, which was subsequently dried with a filter paper. The grids were imaged using a FEI CM 120 electron microscope with a Tietz F416 CMOS camera.

**Circular dichroism.** Five microliter of sample volume were centrifuged for 10 min at 20,000g in an Eppendorf centrifuge 5424. The pellet was resuspended in 60 µL distilled water after discarding the supernatants and transferred to a 0.02 cm pathlength cuvette. CD data were collected from 185 to 260 nm by using a Chirascan-plus qCD spectrometer (Applied Photophysics, Randalls Rd, Leatherhead, UK) at 20 °C, 1.5 time-per-point (s) in 1 nm steps. The datasets were averaged from two repeats. All spectra were baseline corrected against distilled water and smoothed (window size: 8).

**Fluorescent dye binding.** The fluorescent dyes curcumin and FSB were dissolved in DMSO, HS-68 in distilled water at a stock solution of 5 mM. The stock solution was aliquoted and stored at −20 °C in the dark. From the dye stocks working solutions of 50 µM dye in 50 mM HEPES, 100 mM NaCl, pH 7.4 were freshly prepared, the final DMSO concentration for curcumin and FSB was 1%. 100 µL working dye solution was mixed thoroughly with 10 µL of aSyn fibrils to a final protein-to-dye ratio of ~1:6. The samples were transferred to a 96-well microplate (PS, µCLEAR, black, chimney well from Greiner). The fluorescence measurements were performed on a TECAN Spark 20M reader and read from the top at 25 °C with z-Position: 17191 µm, intensity averaged over 30 flashes, 2 nm steps. Curcumin: ex.: 440 nm, gain: 80, ex. bandwidth: 20 nm, em. bandwidth: 20 nm, 485–800 nm. FSB: ex.: 380 nm, gain: 60, ex. bandwidth: 10 nm, em. bandwidth: 10 nm, 450–800 nm. HS-68: ex.: 450 nm, gain: 60, ex. bandwidth: 10 nm, em. bandwidth: 10 nm, 510–800 nm. All spectra were baseline subtracted by dye buffer blank measurements and smoothed (windows size: 8).

**PCA.** For each aSyn fibril sample, fluorescence spectra of the three individual dyes were measured independently twice. The baseline corrected and normalized fluorescent spectra were used as input for PCA. To this end, the curcumin/HS-68 data matrix was generated joining combinatorially the two curcumin spectra (157 data points each) with the two HS-68 spectra (163 data points each) resulting in four datasets per each aSyn fibril. The same procedure was used to generate the curcumin/FSB data matrix. The curcumin/HS-68 as well as the curcumin/FSB matrices were used as input for singular value decomposition[41] using R's prcomp function (stats package, version 3.5.2, zero centering flagged). The results were plotted with R's ggbiplot package (version 0.55).

Similarly, the 99 per-residue protonation levels determined by NMR-based HD exchange measurements of the 25 aSyn fibril samples were used to generate the input data matrix for singular value decomposition using prcomp (with zero centering) and ggbiplot, see above.

**NMR spectroscopy.** HD exchange coupled to NMR was performed according to a previously established protocol[21]. To decrease the rate of HD exchange and thus allow a better definition of regions with small differences in solvent protection, forward HD exchange was done in 0.1% formic acid (pD 4.0) in 99.9% D$_2$O at 37 °C. The forward HD exchange of amide protons in aSyn fibrils was stopped after 1.5 days, fibrils were collected by ultracentrifugation (2 h, 165,000g), frozen in liquid nitrogen and stored at −80 °C. Subsequently, back-exchange in the

dissociation buffer (4M GdSCN, 0.4% formic acid, pH 2.5, in 60% D$_2$O) was followed by 120–140 2D $^1$H-$^{15}$N heteronuclear single quantum coherence (HSQC) experiments (experimental time for each HSQC: 8 min 16 s; number of increments in $^{15}$N dimension: 128) recorded over a time period of ~19 h at a temperature of 3 °C. NMR experiments were measured on a Bruker 800 MHz spectrometer equipped with a 5 mm triple-resonance, pulsed-field z-gradient cryoprobe. Preparative steps such as shimming and temperature equilibration resulted in a dead time of about 7 min (measured for each data set) before the start of the first HSQC. Acquired spectra were processed with linear prediction for the $^{15}$N dimension and analyzed by NMRPipe[42]. Backbone resonance assignments of N-terminally acetylated aSyn in the dissolving buffer (4M GdSCN, 0.4% formic acid, pH 2.5) were obtained with the help of 3D HNCA, HN(CO)CA, HNCACB, CBCA(CO)NH, HNCO and HN(CA)CO experiments[43], in which indirect frequency dimensions were sampled in a non-uniform manner[44].

**Analysis of HD back-exchange.** Previous HD exchange studies conducted on in vitro aggregated aSyn fibrils established that a forward exchange time of 1.5 days allows robust determination of residue-specific protonation levels[21,23]. To further enhance the distinction for residues with different amide proton exchange rates, we used a mixture of H$_2$O and D$_2$O in the dissociation buffer[21]. The relative intensity (height) data for 99 HSQC-separated aSyn residues were extracted from 120–140 2D $^1$H-$^{15}$N HSQC spectra using the centered peak position of the final fully exchanged HSQC spectrum of the series normalized to a value of 1.0 using NMRPipe[42]. For each aSyn residue the back-exchange curves were fitted to a single exponential decay using Prism (Graphpad). Knowing the exact dead time and HSQC experiment duration the relative t$_0$ intensity level was back calculated and the scale was transformed into absolute protonation levels by using the average maximum intensity value of the five residues with highest signal intensity.

**EPR spectroscopy.** aSyn fibril samples were loaded into quartz tubes with O.D. = 1.6 mm and I.D. = 1.1 mm geometry. Room-temperature continuous wave EPR spectra were recorded on a EleXsys E500 spectrometer at X-band (9.8 GHz), equipped with a Super-High Q resonator (ER4122SHQE). Experimental conditions: microwave (MW) power 2 mW, modulation amplitude 5 G, conversion time 10 ms and 150 scans. Continuous wave EPR spectra were simulated using the program 'Chili' in Easyspin software (version 5.1.12)[45]. Two nitroxide species with different weights were used in the simulation. After initial estimation of weights, multiple least-square optimizations were performed, in order to obtain the correlation times and weights for different nitroxide dynamics. For the fitting procedure, g, hyperfine values, peak-to-peak linewidths were kept within a range consistent with previous reports[46].

Q-band (34 GHz) PELDOR measurements were performed on a Bruker EleXsys E580 spectrometer, equipped with a pulsed 170 W Q-band TWT-amplifier (Model 187Ka, Applied Systems Engineering Inc.) and an EN5107D2 EPR/ENDOR probe-head. PELDOR traces were measured using a four-pulse sequence $\left(\frac{\pi^1}{2} - \tau_1 - \pi^1 - (\tau_1 + T) - \pi^2 - (\tau_2 - T) - \pi^1 - \tau_2 - [echo]\right)$ with probe pulse ($^1$) and pump pulse ($^2$)[47]. $T$ was incremented in 8 ns steps. The pump pulse frequency was located at the maximum nitroxide absorption, determined by field-swept echo experiments. The probe pulse frequency was set to 80 MHz lower than the pump frequency (Δν = 80 MHz). The probe and pump pulse lengths were determined by a nutation experiment. The length of the probe π pulse was 26–32 ns, depending on the Q-value of the cavity. The $\frac{\pi}{2}$ pulse length was set to be identical to the length of π pulse by varying the pulse strength or amplitude (w$_1$). This procedure minimizes the overlap between the pump and probe pulse and thus attenuates the so-called '2 + 1' artifact. The π pulse at pump frequency was around 22–26 ns, and $\tau_1$ was set to 200 ns, in order to suppress deuterium modulations.

DeerAnalysis2018[48,49] was used to analyze PELDOR spectra. Background subtraction was performed using 2nd polynomial functions, followed by least-square fitting and Tikhonov regularization with L-curve selection.

**Reporting summary.** Further information on research design is available in the Nature Research Reporting Summary linked to this article.

## Data availability

The data that support the findings of this study are available from the corresponding authors upon reasonable request. The source data underlying Figs. 1b, c, 4a, d, g, 5a and Supplementary Figures 1b and 3c are provided as a Source Data file.

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

## Acknowledgements

The authors thank K. Giller (MPIBPC, Göttingen) and S. Cima (DZNE, Göttingen) for preparation of aSyn samples, K. Overkamp (MPIBPC, Göttingen) for mass spectrometry, and Dr. F. Schmidt and Dr. A. Giese (Ludwig Maximilians University, Munich) for help in the initial phase of the project. HS-68 was prepared by the Facility for Synthetic Chemistry at MPIBPC. Amyloid fibrils were imaged in the Facility for Electron Microscopy at MPIBPC. C.O.F. thanks Agencia Nacional de Promoción Científica y Tecnológica, the Max Planck Society, and the Alexander von Humboldt Foundation for financial support. M.Z. was supported by the Cluster of Excellence DFG Research Center Nanoscale Microscopy and Molecular Physiology of the Brain. S.-J.L. was supported by the National Research Foundation (NRF) grant funded by the Korean Government (MEST) (NRF-2018R1A2A1A05078261 and NRF-2018R1A5A2025964). G.M.H. was supported by the National Health and Medical Research Council of Australia. W.S.K. and G.M.H. thank the MSA Coalition for research funding.

## Author contributions

T.S. conducted protein preparation, fibril seeding, dye binding, CD and NMR data acquisition and analysis; B.C.J. performed brain extract preparation, protein preparation and PMCA; S.-H.L. and M.B. performed EPR spectroscopy and analyzed EPR data; S.B. and C.O.F. contributed reagents; D.R. supervised electron microscopy; G.M.H. diagnosed the human brain tissue samples; G.M.H., W.S.K., S.-J.L. and M.Z. designed the project; T.S., G.M.H., W.S.K., S.-J.L. and M.Z. wrote the paper.

## Competing interests

S.-J.L. is a founder and co-CEO of Neuramedy Co., Ltd. The other authors declare no competing interests.
