## [Peer Review File · Nature Communications]

Reviewers' comments:

Reviewer #1 (Remarks to the Author):

The manuscript "Diversity of a-synuclein aggregate structures in neurodegenerative diseases" by Strohaker et al aims to define how a-synuclein aggregate structure impacts on disease phenotype, using PMCA amplification from brain extracts of patients with Parkinson's disease, Lewy body dementia, or multiple system atrophy in combination with various biophysical methods.

The authors demonstrate conformational differences of human sample-amplified a-synuclein fibrils compared to recombinant a-synuclein fibrils using the dyes HS-68, FSB and curcumin. NMR-based hydrogen-deuterium exchange experiments show varied solvent accessibility of the various a-synuclein fibrils at the N- and C-termini of the protein. Yet, the authors show significant clustering of the three disease-derived fibrils whereas the recombinant a-synuclein fibrils varied substantially from the human samples. Last, the authors use electron paramagnetic resonance and demonstrate no defined distance of their chosen pair of residues within a-synuclein fibrils, due to not defined reasons.

Overall, this manuscript is very timely and novel as the reasons why different a-synucleinopathies exhibit variable pathologies and phenotypes are unknown. Multiple groups have proposed and shown differences in aggregation and toxicity of various strains of a-synuclein fibrils. Yet, so far, there is no structural basis for that and the only studies performed in this direction use recombinant a-synuclein.

The manuscript is written clearly and easy to follow, with straight-forward experiments to address the central questions.

Yet, my biggest concern is the lack of demonstration of reproducibility. The authors should use their PMCA protocol multiple times on the same patient sample to assess variability in the fibril generating process. If this process does not yield a similar fibril structure, then the comparison between samples within PD, MSA and DLB groups, and the comparison between fibrils in different diseases is meaningless. Thus, at this point, the outcome of the study is overstated.

In addition, the following points should be addressed:

Fig. 1b: Where same data obtained for the other patient-derived aggregates? Can input be shown?

Fig. 1b,c: How can the authors ensure that not recombinant aSyn produced the increase in proteinase K-resistant aggregates and ThioT-positive fluorescence? How reliable is PMCA in amplifying existing a-synuclein fibrils?

Fig. 1d: Were the signals normalized to the input amount? The relative differences may be simply due to different burden of a-synuclein aggregates in the respective patient samples.

Fig. 2b: How does buffer alone look like? And can the spectra for HS-68 and FSB also be shown since they are included in panel c? Also, it is unclear what the original material was for the de novo aggregation.

Figs. 2 and Suppl. 5: Is there a reason PD3, MSA5 and DLB2 were excluded from the analysis? Having only 1 DLB patient as comparison is not enough to draw significant conclusions.

Fig. 3a: The authors describe that the three chosen brain-derived profiles were distinctive from one another, but it should be shown how all the other brain-derived samples look like since patient-to-patient variation is expected.

Discussion: The authors write "our data provide a mechanistic explanation for the known clinical variations in a-synucleinopathies". This is too much of a simplification, because different cellular environments as well as different genetic backgrounds in patients likely provide a tremendous impact on a-synuclein fibrillation outcome.

Minor points:

Methods: No details on the ThT fluorescence assay are given.

Online content: The sentence is missing information.

Reviewer #2 (Remarks to the Author):

The EPR data acquisition and analysis appears to have been carried out to the highest standards. The conclusions drawn are consistent with the data.

Reviewer #3 (Remarks to the Author):

The paper by Strohäker et al looks at diversity in alpha-synuclein (AS) structure by dye fluorescence, NMR, and EPR. The correlation between amyloid heterogeneity and pathology is a timely, and increasingly popular theme in the field of aggregative misfolding. The work uses state of the art methods with experiments that appear to be well done. The main problems with the paper are with the interpretation of the data, due to: (1) a small sample size to analyze the diversity of AS heterogeneity as it relates to human disease, (2) principal component analyses (PCAs) that do not support the author's central conclusion that different disease phenotypes are due to differences in AS aggregate structure.

Major:

1) The sample sizes of AS aggregates used to propagate aggregate structures through PCMA is small. PD (n=5), MSA (n=5), DLB (n=2), cited from Fig. 1A. This is understandable, since the template samples for PCMA come from human donor brains. However, the sample sizes may be too small to draw meaningful conclusions on a potentially important medical topic. Moreover, the authors should check if the sample sizes warrant the principal component (PC) analysis (Fig. 2c, 3b, 5S). on which the authors base their conclusions. See for example Shaukat S.S., Rao T.A., Khan M.A.: Impact of sample size on principal component analysis ordination of an environmental data set: effects on eigenstructure. *Ekológia (Bratislava)*, Vol. 35, No. 2, p. 173–190, 2016, and references therein.

2) In the PCA plots of figures Fig. 2c, 3b, 5S, the PD and MSA data (purple and green) overlap. With the DLB patient data, the situation is less clear since there are only 2 samples (orange) and the variance in replicate measurements seems comparable or larger the variance between the two samples. Given that the data from 2 of 3 (and likely all 3) disease types overlap the conclusions of the authors that there is a "structural basis for disease phenotype" seems not to be supported by the data. To some extent this is acknowledged by the authors in lines 75-86 of the text but not explained or elaborated on further. The passage in lines 75-86, which notes overlap between the PD and MSA groups (again DLB conclusions seem tenuous with only 2 samples), seems to contradict the much stronger statements in the abstract and in the closing paragraph (lines 166-172).

The most striking difference (e.g. Fig. 2C) seems to be between aggregates propagated from recombinant AS (red blue, and yellow) and brain extract AS (orange, green, purple). It's unclear why this should be unless the brain extracts are subject to post-translational modification, aging, or some other process that propagates structures different from those in vivo. This potentially interesting question on the difference between recombinant and brain-derived subtypes is not addressed in the MS.

3) What is being compared in the PCA analyses should have been better described:

- a) For the fluorescence data what is being compared? Emission maximum (fluorescence wavelength), quantum yield (fluorescence intensity), or something else?
- b) NMR provides residue-level information with potentially 140 data points on proton occupancy per sample. So what is being compared in the PCA?

It would have been useful if the data variance would have been illustrated in a more straightforward way, for example: uncertainty bars giving the standard error for the NMR proton occupancies between the five PD samples in Fig 3C, and perhaps a panel showing the fluorescence spectra of all 5 PD samples in Fig. 2B instead of just sample PD1.

Why is the variance in EM samples not considered, and what is the variance supposed to be, if any, in Fig. 2A. Shorter fibrils for PD, or is this just for PD1? Arguably EM would be the most direct measure of fibril structure. NMR proton occupancy or fluorescence emission wavelength can depend on factors other than structure, such as solvent/dye accessibility and/or dye binding avidity. These and protease susceptibility, could for example be affected by packing of protofibrils in the fibril, rather than specific changes in the H-bonded structure of the steric zipper beta-sheet.

4) It's surprising that there is not aggregate heterogeneity within a single patient. What is the mechanism to guard against multiple morphologies in the same patient and has this even been looked at? That is, has an analysis of the variance in fibril morphology (by EM) or dye binding properties been looked at within a single patient?

5) I'm not sure why the EPR data in Fig. S9-S11 and Table S1 are included in the supporting information. The EPR data seems like new tangential data (rather than supporting information for data or results in the main text). They are only discussed in one paragraph (lines 126-146) that adds little to the paper and is difficult to peer review based on the short exposition in the paragraph. The EPR data and the paragraph in the main text that refers to them, should probably be removed as it would be difficult to assess in detail by either a reviewer or reader of the paper.

Minor:

1) In Fig. 1C - it is hard to tell which are open circles, because the red circles are covering the open circles for the "no seed" data. It might be better to use a different color (e.g. green) than black open circles.

2) In Supporting Fig. S6 - In A the cross peak labeled are E20, E75, N122. In panel B the exponential decays are shown for E20, T75, A90. I'm not sure why the exponentials could not have been shown for the three residues labeled in panel A. Also the colors in the B panels are often hard to distinguish

Reviewer 1:

The manuscript "Diversity of a-synuclein aggregate structures in neurodegenerative diseases" by Strohaker et al aims to define how a-synuclein aggregate structure impacts on disease phenotype, using PMCA amplification from brain extracts of patients with Parkinson's disease, Lewy body dementia, or multiple system atrophy in combination with various biophysical methods.

The authors demonstrate conformational differences of human sample-amplified a-synuclein fibrils compared to recombinant a-synuclein fibrils using the dyes HS-68, FSB and curcumin. NMR-based hydrogen-deuterium exchange experiments show varied solvent accessibility of the various a-synuclein fibrils at the N- and C-termini of the protein. Yet, the authors show significant clustering of the three disease-derived fibrils whereas the recombinant a-synuclein fibrils varied substantially from the human samples. Last, the authors use electron paramagnetic resonance and demonstrate no defined distance of their chosen pair of residues within a-synuclein fibrils, due to not defined reasons.

Overall, this manuscript is very timely and novel as the reasons why different a-synucleinopathies exhibit variable pathologies and phenotypes are unknown. Multiple groups have proposed and shown differences in aggregation and toxicity of various strains of a-synuclein fibrils. Yet, so far, there is no structural basis for that and the only studies performed in this direction use recombinant a-synuclein.

Reply: We thank the referee for her/his kind words.

The manuscript is written clearly and easy to follow, with straight-forward experiments to address the central questions. Yet, my biggest concern is the lack of demonstration of reproducibility. The authors should use their PMCA protocol multiple times on the same patient sample to assess variability in the fibril generating process. If this process does not yield a similar fibril structure, then the comparison between samples within PD, MSA and DLB groups, and the comparison between fibrils in different diseases is meaningless. Thus, at this point, the outcome of the study is overstated.

Reply: As the referee points out, this is the first structural study, which starts from patient brain extracts and does not only use recombinant a-synuclein. This involved a number of challenges including a sufficient amount of brain extracts from multiple patients with different diseases, controls that exclude *de novo* aggregation during seed amplification, establishment of assays that allow a robust structural analysis of multiple samples, and establishment and calibration of methods, which provide sufficient resolution to distinguish between different amyloid structures. In addition, all steps have to be feasible in a time frame such that the overall analysis can be done in a few years. In addition, a certain level of uncertainty/variation will be associated with each of these steps. We hope that the referee agrees that, with the work reported in the current manuscript, it is probably the first time that these challenges have really been tackled.

With respect to the variations introduced by PMCA, we chose to test this question by using the PMCA protocol multiple times in conditions where brain-extract material is present but *de novo* aggregation occurs because protein concentration and PMCA duration were doubled. We chose this approach, because (i) *de novo* aggregation is often considered to be less reproducible when compared to seeded aggregation, and (ii) to exclude that *de novo* aggregation under PMCA conditions is the reason for the distinct structural properties of brain-extract amplified fibrils when compared to hsAsyn/lsAsyn polymorphs. As expected, some variation in the structural properties as assessed by the dye binding assay and in the HD plot were present (yellow data points in Fig. 3d-e, 5b). Nevertheless, all the "*de novo* PMCA" data cluster in a distinct region of both the dye PCA plot (Fig. 3d-e) and the HD PCA plot (Fig. 5b), i.e. all "*de novo* PMCA" fibrils have similar conformations. In addition, the analysis showed that "*de novo* PMCA" fibrils are distinct from the *in vitro* polymorphs (hsAsyn, lsAsyn) and - even more important - from the brain-extract amplified fibrils (i.e. when *de novo* aggregation did not occur during PMCA). From these data, we believe that it is safe to conclude that any potential variations introduced by PMCA do not preclude the distinction between different

a-synuclein fibril structures (as for example between brain extract-amplified fibrils, hsAsyn and lsAsyn; Fig. 3d-e, 5b).

Please also note that in the dye PCA plot, the spread in PD samples is larger than for MSA samples (Fig. 3d-e). This suggests that most of the variations observed in the PCA plots within a specific patient group are not due to variations inherent to PMCA, because in this case we would expect to observe a similar spread in both patient groups. At the same time, the comparison of the two groups indicates that we would have to repeat at least 5 times the PMCA, in order to reliably estimate the influence of PMCA amplification on the dye spectra of a single patient. In addition, because a different spread was observed in the PD and MSA group, this procedure would have to be repeated for at least two patients (maybe one PD and one MSA patient). Given the large number of controls, which are already part of the manuscript, these additional studies are currently beyond the scope of the manuscript.

What remains to be determined is if, for example, (i) a larger number of patients, or (ii) more stratified patient groups, or (iii) the usage of extracts from a different brain region (in the current study human amygdala was used) would result in narrower distributions of dye binding properties/HD profiles and thus allow a structural distinction between for example PD and MSA (or enable subtyping of PD patients). To better address these issues, we have largely rewritten the Discussion section (pages 13-15 of the revised manuscript).

In addition, the following points should be addressed: Fig. 1b: Where same data obtained for the other patient-derived aggregates? Can input be shown?

Reply: The figure below shows the requested data:

Figure. (a) Immunoblotting of brain extracts from PD, MSA, DLB and control (CT; brain extract from an individual, in which a synucleinopathy was excluded). (b) Immunoblotting of PMCA products with indicated brain extracts as seeds. High molecular weight aSyn was induced through PMCA with brain extracts from patients with PD, MSA or DLB, whereas control brain extract failed to induce high molecular weight aSyn during PMCA. (c) Immunoblotting of PMCA products with proteinase-K (PK) digestion. PK-resistant signals were detected in the PD, MSA and DLB samples, whereas nothing remained in the control sample.

Fig. 1b,c: How can the authors ensure that not recombinant aSyn produced the increase in proteinase K-resistant aggregates and ThioT-positive fluorescence? How reliable is PMCA in amplifying existing a-synuclein fibrils?

Reply: Please note that PMCA with brain material from a control patient did not result in proteinase K-resistant aggregates (Fig. 1a and figure above). In addition, when no brain material was added, i.e. only the recombinant protein was present, no increase in ThT was observed (“no seed” in Fig. 1b).

Setup of the PMCA involved optimization of the concentration of recombinant protein, as well as the number of cycles and the duration of sonication, and strictly required that in the absence of brain extracts no ThT-positive aggregates were formed.

Fig. 1d: Were the signals normalized to the input amount? The relative differences may be simply due to different burden of a-synuclein aggregates in the respective patient samples.

Reply: When PMCA or similar methods such as RT-Quick are used, it is often the differences in aggregation kinetics and ThT intensity at saturation that are analyzed (while not always taking into account differences in the burden of a-synuclein aggregates). In the current study, the data shown in Fig. 1c just serve to demonstrate that PMCA successfully amplified ThT-positive aggregates, i.e. amyloid fibrils, without any attempt to compare the “efficiency” of aggregation between different patient samples. The signals shown in Fig. 1c were therefore not normalized. Indeed, we are convinced that it is essential to pursue a structural approach (as followed in the current study), in order to minimize the influence of variations in a-synuclein aggregate concentrations and thus provide reliable insight into a-synuclein strains.

Fig. 2b: How does buffer alone look like? And can the spectra for HS-68 and FSB also be shown since they are included in panel c? Also, it is unclear what the original material was for the de novo aggregation.

Reply: The spectra of HS-68 and FSB are now shown in Fig. 3. All fluorescence “spectra were baseline subtracted by dye buffer blank measurements and smoothed (windows size: 8)” (stated in the revised method section).

In the revised version of the manuscript, we now describe in detail the different *in vitro* polymorphs including the *de novo* PMCA fibrils (page 4-5):

“In addition, we prepared different aSyn fibrils not specific for individual patients (termed in vitro polymorphs). One in vitro polymorph was generated by adding brain extracts from different patients to the PMCA reaction, but doubling both the concentration of recombinant aSyn and the PMCA duration. Because of the higher aSyn concentration and the longer PMCA duration, de novo aggregation of aSyn occurs. The resulting aSyn fibrils serve as an important control, because they represent an in vitro polymorph, which was generated under exactly the same buffer conditions as the fibrils amplified from brain extracts.

Two more aSyn polymorphs were produced through aggregation of recombinant aSyn in the absence of PMCA-seeds under two different buffer conditions: 50 mM Tris-HCl, 150 mM KCl, pH 7.5, 0.02 % NaN₃ (termed “de novo hsAsyn”), and 5 mM Tris-HCl, pH 7.5, 0.02 % NaN₃ (termed “de novo lsAsyn”)^{5,6}. Previous studies showed that de novo hsAsyn and de novo lsAsyn differ in their fibril structure and also in their toxicity when added to cells or injected into animal models of aSyn-diseases^{5,6}. In addition, de novo hsAsyn and de novo lsAsyn fibrils were used for seeding of aSyn aggregation, in order to investigate the influence of the seeding reaction on aSyn fibril structure. These fibrils were termed “hsAsyn” and “lsAsyn”. Circular dichroism and electron microscopy showed that hsAsyn and lsAsyn are amyloid fibrils with different morphologies (Fig. 2).

Finally, aSyn was aggregated de novo in 50 mM HEPES, 100 mM NaCl, pH 7.4, 0.02 % NaN₃ and subsequent seeded aggregation in the same condition. This provided yet another in vitro polymorph (termed “high salt [HEPES]”), which can be compared to aSyn fibrils amplified from brain extracts.”

and on page 7:

“The de novo PMCA fibrils (yellow data points in Fig. 3d-e), i.e. the fibrils that were obtained through de novo aggregation in the presence of brain extracts (PD1, PD2, PD4, PD5, MSA1, MSA2, MSA3 and MSA4 according to Table 1), also formed a single cluster, which did not overlap with either the hsAsyn or the lsAsyn cluster (Fig. 3d-e).”

Figs. 2 and Suppl. 5: Is there a reason PD3, MSA5 and DLB2 were excluded from the analysis? Having only 1 DLB patient as comparison is not enough to draw significant conclusions.

Reply: We agree with the referee that the number of DLB patients is currently too low. In the revised version of the manuscript, we therefore focus on the two “major” synucleinopathies PD and MSA

and removed the DLB data (*Prusiner et al. PNAS 2015* also concentrated on PD and MSA). Hopefully we will get material from more DLB patients in the future, to study a-synuclein aggregate structures in DLB. As far as PD3 and MSA5 in Fig. 3 (previously Fig. 2) are concerned, this is due to the “history” of the project, where we started with the dye binding assay and then went to HD exchange. At that time we managed to get samples for another PD and MSA patient, which we then subjected to HD exchange.

Fig. 3a: The authors describe that the three chosen brain-derived profiles were distinctive from one another, but it should be shown how all the other brain-derived samples look like since patient-to-patient variation is expected.

Reply: Thanks for the suggestion. All brain-derived profiles are shown in Fig. 5a of the revised manuscript. The variations in these profiles are the reason for the spread in the PCA plot (Fig. 5b) and reflect the patient-to-patient variation. We stress in the revised version that the profiles shown in Fig. 6 (previously Fig. 3a) are the most distinct/extreme cases (page 11):

“In agreement with the observed residue-specific differences in the protonation levels (Fig. 5a), PD1 and PD2 were most distinct from MSA1 and MSA2 (Fig. 5b).

To gain insight into the conformational basis of this distinction, we compared the HD profile averaged over PD1 and PD2 with that averaged over MSA1 and MSA2 (Fig. 6).”

Discussion: The authors write “our data provide a mechanistic explanation for the known clinical variations in a-synucleinopathies”. This is too much of a simplification, because different cellular environments as well as different genetic backgrounds in patients likely provide a tremendous impact on a-synuclein fibrillation outcome.

Reply: We fully agree. We removed this statement and largely extended the discussion, in order to better put our findings into the proper context (page 13-15 of the revised manuscript):

Minor points: Methods: No details on the ThT fluorescence assay are given.

Reply: Included in the revised version.

Reviewer 2:

The EPR data acquisition and analysis appears to have been carried out to the highest standards. The conclusions drawn are consistent with the data.

Reply: We thank the referee for her/his kind words.

Reviewer 3:

The paper by Strohäker et al looks at diversity in alpha-synuclein (AS) structure by dye fluorescence, NMR, and EPR. The correlation between amyloid heterogeneity and pathology is a timely, and increasingly popular theme in the field of aggregative misfolding. The work uses state of the art methods with experiments that appear to be well done.

Reply: We thank the referee for her/his comments.

The main problems with the paper are with the interpretation of the data, due to: (1) a small sample size to analyze the diversity of AS heterogeneity as it relates to human disease, (2) principal component analyses (PCAs) that do not support the author’s central conclusion that different disease phenotypes are due to differences in AS aggregate structure.

Reply: The small sample size was determined by the number of MSA and PD patient samples, which was available to us, i.e. the need to obtain human donor brains. Please note that our sample size is

comparable to patient numbers used in other studies investigating AS strains. **For example in Peng et al. Nature 2018, the work was based on 6 MSA (including two different MSA subtypes; i.e. 4 MSA-P and 2 MSA-C), 6 PDD and 1 DLB patient** (Extended Data Table 1 in Peng et al. Nature 2018). To decrease the influence of small sample size on data interpretation, we deleted the DLB data from the revised manuscript, i.e. focus now on the two “major” synucleinopathies PD and MSA (a similar focus was chosen in Prusiner et al. PNAS 2015), for which we had 5 patient samples each.

We apologize if the presentation of our data/experiments was unclear. In the revised version of the manuscript, we describe in more detail the PCA analyses. The PCA analyses were done for all aSyn fibrils together (i.e. both *in vitro* polymorphs and aSyn fibrils amplified from brain extracts) and using the full experimental data sets (i.e. the full digitized/binned fluorescence spectra in case of the dyes; all residue-specific HD protonation levels in case of NMR). Thus, we were working with high-dimensional data sets (for example in case of HD exchange: 99 protonation values for non-overlapping aSyn cross-peaks x 25 aSyn fibril types; please see Fig. 5b), which were transformed by PCA.

In addition, we believe that a short coming of the originally submitted manuscript related to the wording of the abstract and the concluding paragraph. **In the revised manuscript, we modified the abstract and extended the discussion, highlighting the currently available evidence for different AS aggregate structures in different diseases and stressing our major findings:**

- a significant range/diversity in the structure of AS aggregates within a single disease phenotype, as illustrated by the variations in the dye binding properties (Fig. 3) and the variations in the HD exchange profiles (Fig. 5),
- that the structures of AS aggregates amplified from patient brain extracts differ from the structure of the *in vitro* aggregates hsAsyn/lsAsyn, which are/were used to investigate aspects of synucleinopathies in cell and animal models in many labs worldwide (for a review, please see Alam et al. J Neurochem 2019).

Please find below a detailed response made to improve our work.

1) The sample sizes of AS aggregates used to propagate aggregate structures through PCMA is small. PD (n=5), MSA (n=5), DLB (n=2), cited from Fig. 1A. This is understandable, since the template samples for PMCA come from human donor brains. However, the sample sizes may be too small to draw meaningful conclusions on a potentially important medical topic. Moreover, the authors should check if the sample sizes warrant the principal component (PC) analysis (Fig. 2c, 3b, 5S). on which the authors base their conclusions. See for example Shaukat S.S., Rao T.A., Khan M.A.: Impact of sample size on principal component analysis ordination of an environmental data set: effects on eigenstructure. *Ekológia (Bratislava)*, Vol. 35, No. 2, p. 173–190, 2016, and references therein.

Reply: Please note that the number of MSA and PD patient samples used in the current work is in the range used in other recent studies investigating AS strains. For example in Peng et al. Nature 2018, the work is based on 6 MSA (including two different MSA subtypes; i.e. 4 MSA-P and 2 MSA-C), 6 PDD and 1 DLB patient (Extended Data Table 1 in Peng et al. Nature 2018). These quite small sample sizes are related to the need to obtain human donor brains. To decrease the influence of small sample size on data interpretation, we focus now on the two “major” synucleinopathies PD and MSA, for which we had 5 patient samples each, and deleted the DLB data from the revised manuscript.

Regarding the PCA analysis, we apologize that we did not provide sufficient details in the originally submitted manuscript. In the revised manuscript, the PCA analyses are described on pages 6-7:

“The above data show that both the shape and the peak wavelength of the fluorescence spectra of curcumin, HS-68 and FSB change for different aSyn fibrils. To fully take advantage of this sensitivity and increase the discriminative power, we performed principal component analysis (PCA). To this end, the normalized fluorescence spectra of two dyes for a specific aSyn fibril type were concatenated and treated as a single spectrum. Because curcumin fluorescence spectra showed the largest diversity, we combined the buffer baseline corrected and normalized fluorescence spectra of curcumin with either HS-68 or FSB. The data points from these artificial

spectra (curcumin + HS-68: 157 + 163 data points; curcumin + FSB: 157 + 166 data points) of the different aSyn fibrils were subsequently analyzed by PCA (Fig. 3d-e). The two largest principal components (PCs) accounted for more than 90% of the variations in the spectra of curcumin, HS-68 and FSB.”

and on page 11:

“In Fig. 4 and 5, H/D protonation levels of 99 non-overlapping residues in more than 20 aSyn fibril samples are shown. To allow an unbiased comparison of all protonation levels, this high-dimensional data set was transformed using PCA (Fig. 5b). The two largest principal components accounted for 76.4% of the variations in the HD data and clustered the de novo PMCA fibrils in a small region of the PCA plot (Fig. 5b).”

i.e. both in the case of the fluorescence spectra and the HD data we were working with high dimensional data sets that were transformed by PCA. Please also note that in the revised manuscript the PCA analyses were redone without the DLB data.

2) In the PCA plots of figures Fig. 2c, 3b, 5S, the PD and MSA data (purple and green) overlap. With the DLB patient data, the situation is less clear since there are only 2 samples (orange) and the variance in replicate measurements seems comparable or larger the variance between the two samples. Given that the data from 2 of 3 (and likely all 3) disease types overlap the conclusions of the authors that there is a “structural basis for disease phenotype” seems not to be supported by the data. To some extent this is acknowledged by the authors in lines 75-86 of the text but not explained or elaborated on further. The passage in lines 75-86, which notes overlap between the PD and MSA groups (again DLB conclusions seem tenuous with only 2 samples), seems to contradict the much stronger statements in the abstract and in the closing paragraph (lines 166-172).

Reply: We apologize if the presentation of our data was misleading. In the revised version of the manuscript, we changed the abstract and extended the discussion, highlighting the currently available evidence for different AS aggregate structures in different diseases and stressing our major findings (pages 13-15 of the revised manuscript). In addition, we removed the DLB data, because we only had access to material from 2 patients.

The most striking difference (e.g. Fig. 2C) seems to be between aggregates propagated from recombinant AS (red blue, and yellow) and brain extract AS (orange, green, purple). It’s unclear why this should be unless the brain extracts are subject to post-translational modification, aging, or some other process that propagates structures different from those in vivo. This potentially interesting question on the difference between recombinant and brain-derived subtypes is not addressed in the MS.

Reply: Thanks for stressing this point. We fully agree that this is a major result from our work, which we now stress in the abstract and the discussion section (pages 13-15) of the revised manuscript. Indeed, in the case of aggregates/fibrils of the protein tau, which is found in insoluble deposits in patients with Alzheimer’s disease and a number of other neurodegenerative diseases, cryo-EM showed that the structure of *in vivo* Tau aggregates differs from Tau fibrils aggregated *in vitro* (Fitzpatrick et al. *Nature* 2017; Falcon et al. *Nature* 2018; Falcon et al. *Nature* 2019; Zhang et al. *eLife* 2019). Although these data are based on a small number of patients, they highlight the potential differences between *in vitro* and *in vivo* aggregates. Potential reasons for these differences are currently unknown, but could involve post-translational modifications or the presence of co-factors. Consistent with the later hypothesis, additional/unresolved electron density in the cryo-EM structure of tau fibrils from chronic traumatic encephalopathy was attributed to an unknown hydrophobic molecule (Falcon et al. “Novel tau filament fold in chronic traumatic encephalopathy encloses hydrophobic molecules” *Nature* 2019).

3) What is being compared in the PCA analyses should have been better described:

- a) For the fluorescence data what is being compared? Emission maximum (fluorescence wavelength), quantum yield (fluorescence intensity), or something else?
- b) NMR provides residue-level information with potentially 140 data points on proton occupancy per sample. So what is being compared in the PCA?

Reply: In the revised version of the manuscript, the PCA analyses are described on pages 6-7:

“The above data show that both the shape and the peak wavelength of the fluorescence spectra of curcumin, HS-68 and FSB change for different aSyn fibrils. To fully take advantage of this sensitivity and increase the discriminative power, we performed principal component analysis (PCA). To this end, the normalized fluorescence spectra of two dyes for a specific aSyn fibril type were concatenated and treated as a single spectrum. Because curcumin fluorescence spectra showed the largest diversity, we combined the buffer baseline corrected and normalized fluorescence spectra of curcumin with either HS-68 or FSB. The data points from these artificial spectra (curcumin + HS-68: 157 + 163 data points; curcumin + FSB: 157 + 166 data points) of the different aSyn fibrils were subsequently analyzed by PCA (Fig. 3d-e). The two largest principal components (PCs) accounted for more than 90% of the variations in the spectra of curcumin, HS-68 and FSB.”

and on page 11:

“In Fig. 4 and 5, H/D protonation levels of 99 non-overlapping residues in more than 20 aSyn fibril samples are shown. To allow an unbiased comparison of all protonation levels, this high-dimensional data set was transformed using PCA (Fig. 5b). The two largest principal components accounted for 76.4% of the variations in the HD data and clustered the de novo PMCA fibrils in a small region of the PCA plot (Fig. 5b).”

and in the methods section on page 19:

“PCA

For each aSyn fibril sample, fluorescence spectra of the three individual dyes were measured independently twice. The baseline corrected and normalized fluorescent spectra were used as input for PCA. To this end, the curcumin/HS-68 data matrix was generated joining combinatorially the two curcumin spectra (157 data points each) with the two HS-68 spectra (163 data points each) resulting in four datasets per each aSyn fibril. The same procedure was used to generate the curcumin/FSB data matrix. The curcumin/HS-68 as well as the curcumin/FSB matrices were used as input for singular value decomposition⁴¹ using R's `prcomp` function (`stats` package, version 3.5.2, zero centering flagged). The results were plotted with R's `ggbiplot` package (version 0.55).

Similarly, the 99 per-residue protonation levels determined by NMR-based HD exchange measurements of the 25 aSyn fibril samples were used to generate the input data matrix for singular value decomposition using `prcomp` (with zero centering) and `ggbiplot`, see above.”

Please note that PCA analyses were redone without the DLB data for the revised manuscript.

It would have been useful if the data variance would have been illustrated in a more straightforward way, for example: uncertainty bars giving the standard error for the NMR proton occupancies between the five PD samples in Fig 3C, and perhaps a panel showing the fluorescence spectra of all 5 PD samples in Fig. 2B instead of just sample PD1.

Reply: In the revised version of the manuscript we show the fluorescence spectra and HD profiles of the individual patient samples (new Figs 3,5). In addition, the HD profiles of different *in vitro* polymorphs are now shown in Fig. 4. The new Fig. 5a also includes individual HD exchange profiles of *de novo* aggregation under conditions of PMCA (i.e. when protein concentration and PMCA duration were doubled and thus *de novo* aggregation induced), as well as average values with standard error (right column, lowest row). In case of patient-associated HD profiles, we show in the new Fig. 6 the residue-specific differences (with error bars) between the average NMR proton occupancies of PD1/PD2 and MSA1/MSA2, because PD1 proton occupancies are very similar to PD2 (same for MSA1 and MSA2; please see Fig. 5a). Please note that the samples are from individual patients (with each having a different disease history), therefore it might be misleading to average over all five PD samples.

We are convinced that the PCA analysis is powerful, because it allows a global comparison of fluorescence spectra of multiple dyes (or HD exchange profiles) of different patient samples without the need to compare every single wavelength of the different fluorescence dyes or each single residue in the HD profile.

Why is the variance in EM samples not considered, and what is the variance supposed to be, if any, in Fig. 2A. Shorter fibrils for PD, or is this just for PD1? Arguably EM would be the most direct measure of fibril structure. NMR proton occupancy or fluorescence emission wavelength can depend on factors other than structure, such as solvent/dye accessibility and/or dye binding

avidity. These and protease susceptibility, could for example be affected by packing of protofibrils in the fibril, rather than specific changes in the H-bonded structure of the steric zipper beta-sheet. 4) It's surprising that there is not aggregate heterogeneity within a single patient. What is the mechanism to guard against multiple morphologies in the same patient and has this even been looked at? That is, has an analysis of the variance in fibril morphology (by EM) or dye binding properties been looked at within a single patient?

Reply: In the EM micrographs of the different AS fibrils (both pure *in vitro* fibrils as well as patient extract-amplified fibrils) we observed a certain level of heterogeneity. Below we highlight this in the EM micrographs of PD1 and MSA1. For both samples, more twisted and more straight fibrils are present. In case of MSA1, the bundled straight fibrils are clearly distinct in terms of macroscopic morphology from the twisted fibrils (both marked by arrows).

PD1

MSA1

It is however difficult to quantify the level of heterogeneity, because it is influenced by the variable sticking of fibrils to different parts of the EM grid. In addition, previous studies indicated that there can be a change in twisting even along a single amyloid fibril (Zhang *et al. eLife* 2019). This apparently opens the door for misinterpretation, i.e. the claim for differences in AS aggregate structure, when not considering the heterogeneity in EM micrographs/morphology.

Because fluorescence, NMR and EPR spectroscopy detect all proteins in the sample and not just a sub-ensemble, they characterize the full spectrum of fibril conformations present in each sample. They might therefore be less “sensitive”, but are on the other hand more reliable/conservative probes for differences in AS fibril structure. Despite these inherent limitations, we find consistent differences between the structural properties of the *in vitro* polymorphs (e.g. hsAsyn and lsAsyn) and the AS aggregates amplified from patient brain extracts. The strength of the spectroscopic approach is stressed on page 13 of the revised manuscript:

“The conformational properties of the aSyn fibrils were analyzed using fluorescent probes (Fig. 3), NMR spectroscopy (Fig. 4-6) and electron paramagnetic resonance (Fig. 7). The strength of the spectroscopic approaches is that all molecules contribute to the measurement. They thus characterize the full spectrum of fibril conformations present in each sample. This is important, because amyloid fibrils are often polymorphic²⁸⁻³⁰. The chosen spectroscopic approach is therefore complementary to cryo-electron microscopy of amyloid fibrils purified from patient brain.”

In the current study, human amygdalas of patients were used. Thus, it could be possible that other AS aggregate structures exist in other areas of the brain (now stressed on page 15 of the revised manuscript):

“The most controversial finding from our study is probably that aSyn fibrils propagated from PD and MSA patients do not have clearly distinct structural properties (Fig. 3,5). This observation is in conflict with a one disease-one strain hypothesis, i.e. a unique connection between the clinical disease presentation and a single, defined structure of aSyn fibrils. Currently, the connection between aSyn disease type and aSyn structure is based on a number of indirect observations, including differences in macroscopic morphology of aSyn inclusions in patients², differences in seeding potential/kinetics of material from different synucleinopathies^{4,36}, differences in antibody binding to different aSyn aggregates (e.g. ³), the ability to produce in vitro different aSyn fibril structures^{5,6}. These observations could, however, also be caused by other factors besides differences in aSyn aggregate structure, such as differences in cellular environment, e.g. in PD, aSyn aggregates occur predominantly in neurons, while in MSA they are found in oligodendrocytes², differences in genetic backgrounds in patients, different burden of aSyn aggregates in different diseases/patient samples, as well as differences in post-translational modifications of aSyn aggregates. The macroscopic morphology of aSyn aggregates in patients is also on a different length scale than the molecular structure of amyloid fibrils. In addition, aSyn-rich aggregates in patients not only contain aSyn fibrils but many other components³⁷, which potentially influence both the morphology of insoluble deposits and their potential for seeding pathogenic aggregation. Another aspect that might require attention is the brain region from which aSyn aggregates are extracted. In the current study, human amygdalas of patients were used. Thus, it could be possible that different aSyn aggregate structures exist in different areas of the brain.”

Variations in AS aggregate structure in different brain regions is a highly interesting question, but is outside the scope of the current study (and also requires the availability of the corresponding material from human donors).

5) I'm not sure why the EPR data in Fig. S9-S11 and Table S1 are included in the supporting information. The EPR data seems like new tangential data (rather than supporting information for data or results in the main text). They are only discussed in one paragraph (lines 126-146) that adds little to the paper and is difficult to peer review based on the short exposition in the paragraph. The EPR data and the paragraph in the main text that refers to them, should probably be removed as it would be difficult to assess in detail by either a reviewer or reader of the paper.

Reply: In the revised manuscript, the EPR data were moved to the main text and the description was extended (pages 12-13 and Fig. 7). The EPR data provide independent experimental evidence for the differences in the structure of *in vitro* polymorphs and brain extract-amplified fibrils. Experimental details are given in the method section.

Minor: 1) In Fig. 1C - it is hard to tell which are open circles, because the red circles are covering the open circles for the “no seed” data. It might be better to use a different color (e.g. green) than black open circles.

Reply: Thanks. We changed it as suggested.

2) In Supporting Fig. S6 - In A the cross peak labeled are E20, E75, N122. In panel B the exponential decays are shown for E20, T75, A90. I'm not sure why the exponentials could not have been shown for the three residues labeled in panel A. Also the colors in the B panels are often hard to distinguish.

Reply: Additional cross peaks are shown in the revised version of Fig. S3 (previously Fig. S6).

REVIEWERS' COMMENTS:

Reviewer #1 (Remarks to the Author):

The authors have carefully addressed most of my concerns in their revised manuscript.

Yet, my main concern remains and is still the reproducibility of fibril types from a single patient sample. I understand this is the first structural study from patient samples, which is very exciting, and can appreciate the difficulties associated with such studies. However, especially for a first study, it is vital to demonstrate that same fibril structure can be obtained from the same patient sample using the PMCA protocol and not by "de novo PMCA". I recognize the controls the authors have performed to indirectly provide confidence in their findings, but it would be much more convincing to show this directly.

Minor point: The authors have provided data for all samples for Fig. 1b in their rebuttal. Why not include it in the actual manuscript figure?

Reviewer #3 (Remarks to the Author):

The MS by Strohäcker describing heterogeneity in aSyn samples propagated from patients and in vitro samples has been almost completely re-written in this revision. This has resulted in a much improved, clearer, more rigorous, and robust paper! All of my concerns have been addressed in the author rebuttal. I am still a little worried about the issue of drawing conclusions from a small sample size but I appreciate the extreme difficulties in obtaining samples from human donors. Moreover as the authors point out, work reporting on similarly small sample sizes has been published in Nature journals, for comparison. Overall this revision of the MS is a huge improvement!

Minor point:

For clarity, "hsAsyn" and "lsAsyn" should be explicitly defined as high-salt and low-salt when they are first introduced in the MS text.

Andrei T. Alexandrescu

Reviewer 1:

The authors have carefully addressed most of my concerns in their revised manuscript.

Reply: We thank the referee for her/his kind words.

Yet, my main concern remains and is still the reproducibility of fibril types from a single patient sample. I understand this is the first structural study from patient samples, which is very exciting, and can appreciate the difficulties associated with such studies. However, especially for a first study, it is vital to demonstrate that same fibril structure can be obtained from the same patient sample using the PMCA protocol and not by “de novo PMCA”. I recognize the controls the authors have performed to indirectly provide confidence in their findings, but it would be much more convincing to show this directly.

Reply: We would like to point out that we already included a large number of controls, which proof the high reproducibility of the data and analysis. Indeed, we are convinced that it is fair to state that the number of controls shown in our manuscript is far beyond what has been reported in any other study investigating structural properties of amyloid fibrils (even using methods with much lower resolution).

Minor point: The authors have provided data for all samples for Fig. 1b in their rebuttal. Why not include it in the actual manuscript figure?

Reply: Thanks for the suggestion. We included it as new SI Fig. 1.

Reviewer 3:

The MS by Strohäcker describing heterogeneity in aSyn samples propagated from patients and in vitro samples has been almost completely re-written in this revision. This has resulted in a much improved, clearer, more rigorous, and robust paper! All of my concerns have been addressed in the author rebuttal. I am still a little worried about the issue of drawing conclusions from a small sample size but I appreciate the extreme difficulties in obtaining samples from human donors. Moreover as the authors point out, work reporting on similarly small sample sizes has been published in Nature journals, for comparison. Overall this revision of the MS is a huge improvement!

Reply: We thank the referee for his kind words.

Minor point:

For clarity, "hsAsyn" and "lsAsyn" should be explicitly defined as high-salt and low-salt when they are first introduced in the MS text.

Reply: This is now defined.